# The *Shigella* kinase effector OspG modulates host ubiquitin signaling to escape septin-cage entrapment

Wei Xian [1,8], Jiaqi Fu [2,8], Qinxin Zhang[1], Chuang Li[3], Yan-Bo Zhao[4], Zhiheng Tang[1], Yi Yuan[1], Ying Wang[1], Yan Zhou [5], Peter S. Brzoic[6], Ning Zheng [7], Songying Ouyang [4], Zhao-qing Luo [3]✉ & Xiaoyun Liu [1]✉

*Shigella flexneri* is a Gram-negative bacterium causing severe bloody dysentery. Its pathogenesis is largely dictated by a plasmid-encoded type III secretion system (T3SS) and its associated effectors. Among these, the effector OspG has been shown to bind to the ubiquitin conjugation machinery (E2~Ub) to activate its kinase activity. However, the cellular targets of OspG remain elusive despite years of extensive efforts. Here we show by unbiased phosphoproteomics that a major target of OspG is CAND1, a regulatory protein controlling the assembly of cullin-RING ubiquitin ligases (CRLs). CAND1 phosphorylation weakens its interaction with cullins, which is expected to impact a large panel of CRL E3s. Indeed, global ubiquitome profiling reveals marked changes in the ubiquitination landscape when OspG is introduced. Notably, OspG promotes ubiquitination of a class of cytoskeletal proteins called septins, thereby inhibiting formation of cage-like structures encircling cytosolic bacteria. Overall, we demonstrate that pathogens have evolved an elaborate strategy to modulate host ubiquitin signaling to evade septin-cage entrapment.

*Shigella flexneri* is a human-adapted pathogen that colonizes the intestinal epithelium[1]. Upon ingestion via the fecal-oral route, *S. flexneri* traverses the small intestine to the colon and rectum and crosses the epithelial barrier through M cells. Then the bacteria invade the epithelium at the basolateral face, multiply within the host cell cytoplasm and subsequently spread to neighboring epithelial cells[2]. Successful *S. flexneri* infection is largely dependent on a 200 kb virulence plasmid that encodes a type III secretion system (T3SS) and its effectors[2,3]. These

bacterial effectors, upon delivery into host cells, have the capacity to modulate a wide variety of cellular processes such as cytoskeletal remodeling, cell death and innate immune responses[4–7]. One of the effectors, OspG, has been shown to stably associate with ubiquitin-conjugating enzymes (E2s) and ubiquitin to activate its kinase activity[8–12]. Additional data indicate that this effector may interfere with host inflammatory responses by inhibiting NF-κB activation[8,9]. However, years later the cellular targets of OspG still remain undetermined.

[1]Department of Microbiology and Infectious Disease Center, NHC Key Laboratory of Medical Immunology, School of Basic Medical Sciences, Peking University Health Science Center, 100191 Beijing, China. [2]Department of Respiratory Medicine, Center for Infectious Diseases and Pathogen Biology, Key Laboratory of Organ Regeneration and Transplantation of the Ministry of Education, State Key Laboratory for Diagnosis and Treatment of Severe Zoonotic Infectious Diseases, The First Hospital of Jilin University, 130021 Changchun, China. [3]Department of Biological Sciences, Purdue University, West Lafayette, IN 47907, USA. [4]Key Laboratory of Microbial Pathogenesis and Interventions of Fujian Province University, the Key Laboratory of Innate Immune Biology of Fujian Province, Biomedical Research Center of South China, Key Laboratory of OptoElectronic Science and Technology for Medicine of the Ministry of Education, College of Life Sciences, Fujian Normal University, Fuzhou, China. [5]Institute of Microbiology, College of Life Sciences, Zhejiang University, 310058 Hangzhou, China. [6]Department of Biochemistry, University of Washington, Seattle, WA 98195, USA. [7]Department of Pharmacology, University of Washington, Seattle, WA 98195, USA. [8]These authors contributed equally: Wei Xian, Jiaqi Fu. ✉e-mail: luoz@purdue.edu; xiaoyun.liu@bjmu.edu.cn

Ubiquitination regulates virtually all cellular processes in eukaryotes in which E3 ubiquitin ligases dictate substrate specificity[13]. Among the largest class of E3s, the SCF (Skp1-Cul1-F-box) complex consists of the RING domain protein Rbx1, the adaptor Skp1, the scaffold protein Cul1, and an F-box protein (FBP)[14–16]. As the substrate recruitment module, various FBPs can be dynamically assembled into the SCF to allow the recognition of a vast array of substrates[17–19]. CAND1 is a crucial regulator of the SCF complex by inducing the exchange of FBPs, but whether this process is further regulated by other mechanisms is unknown[20–24].

Herein, we utilized a large-scale phosphoproteomic approach that measures host protein phosphorylation levels upon the introduction of bacterial effectors. This unbiased proteomic screen identified CAND1 as a potential substrate of OspG. Phosphorylation of CAND1 disrupts its interaction with Cul1, which impacts the composition of SCF complexes. Subsequent ubiquitome profiling revealed that septins associated with actin filaments and microtubules were more extensively ubiquitinated in response to OspG, thereby inhibiting septin-cage formation. Our study demonstrates a mechanism by which bacterial pathogens covalently modify a master regulatory protein controlling the host E3 network in order to reshape the ubiquitome profile to escape from host septin entrapment.

## Results

### CAND1 is a kinase substrate of the *Shigella* type III effector OspG

Although the kinase activity of the *S. flexneri* effector OspG has been recognized for many years, its host target remains elusive. We tackled this problem by performing large-scale comparative phosphoproteomics of host cells expressing OspG or its catalytic mutant OspG K53A (Fig. 1a). Candidate substrates are expected to have higher phosphorylation levels in cells producing a functional kinase. In total, we detected 4307 phosphoproteins from 18613 phosphopeptide assignments in four biological replicates (Supplementary Data 1). Of this comprehensive dataset, the single most prominent hit was assigned by a phosphopeptide from CAND1 (cullin-associated NEDD8-dissociated protein 1). The phosphorylated peptide VIRPLDQPSSF-DATPYIK was abundantly detected in cells expressing OspG but not OspG K53A, exhibiting the greatest difference of intensities (Fig. 1b). Further tandem MS (MS/MS) analysis under different dissociation modes unambiguously mapped the phosphorylation site to S558 (Fig. 1c, and Supplementary Fig. 1d).

### OspG promotes CAND1 phosphorylation by a mechanism dependent on OspG-E2/ubiquitin interactions

To verify our findings, we co-expressed Flag-CAND1 in 293T cells together with OspG or its variant OspG K53A. LC-MS analyses of immunoprecipitated CAND1 detected robust signals of the phosphopeptide VIRPLDQPSpS$_{558}$FDATPYIK in cells co-expressing OspG but not OspG K53A (Fig. 1d). In contrast, an unmodified reference peptide LTLIDPETLLPR was measured with comparable MS signals across all samples (Fig. 1d). As negative controls, such signals were absent in cells expressing CAND1 only. Moreover, CAND1 was not phosphorylated in cells transfected with the enteropathogenic *E. coli* (EPEC) effector NleH1, a homolog of OspG with 30% sequence similarity[25] (Fig. 1h). Together, these results demonstrate that OspG specifically promotes CAND1 phosphorylation in a manner dependent on its kinase activity. We further investigated whether OspG interacts with CAND1 by GST pull-down assays. Indeed, purified GST-OspG, but not GST alone, was able to pull down CAND1 after incubation with cell lysates (Fig. 1e), indicating the association of CAND1 with the effector.

Given its role in regulating cullin-RING ubiquitin ligases (CRLs)[21–23], CAND1 is by definition associated with host ubiquitination machinery. OspG activates its kinase activity by physically interacting with E2-Ub conjugates (as well as E2 or Ub)[8,10,11], while its homolog effector NleH1 is incapable of such protein binding[12]. We reasoned that

OspG-E2-Ub interactions may contribute to CAND1 phosphorylation. Furthermore, stable OspG-E2-Ub association may target the effector to the large ubiquitination complexes, where CAND1 is in proximity leading to its modification. To test this hypothesis, we took advantage of two reported OspG variants L99R/P102E (OspG LR/PE) and L190D/L191D (OspG LL/DD) that are unable to bind to UbcH7 and Ub, respectively[9,10]. Recombinantly purified GST-OspG variants pulled down CAND1 as efficiently as the WT effector (Supplementary Fig. 1a). Therefore, OspG likely associates with CAND1 in a manner independent of its interaction with E2-Ub conjugates. When expressed in 293T cells, however, these two mutants (LR/PE and LL/DD) failed to promote CAND1 phosphorylation (Fig. 1d). Together, these data suggest that OspG-E2-Ub interactions are necessary for CAND1 phosphorylation in a mechanism likely by activating the kinase activity of the effector as previously reported[9,10].

### CAND1 is phosphorylated by OspG in vitro and secreted effectors during infection

To exclude potential phosphorylation cascades contributing to CAND1 modification, we sought to determine whether OspG could directly modify CAND1 in vitro. Recombinant OspG and CAND1 proteins purified from *E. coli* cells were incubated in reactions containing ATP. We did not see detectable levels of CAND1 phosphorylation under these conditions, presumably due to low kinase activity of OspG in vitro as previously reported[9–11]. As ubiquitin stimulates its kinase activity via direct binding[9], we added an equal molar of Ub into the reactions and were able to detect robust signals of phosphorylated CAND1 (pCAND1) and self-modified OspG (Fig. 1f, Supplementary Fig. 1b). These data indicate CAND1 as a direct substrate of OspG kinase activity.

Next we investigated whether CAND1 could be modified during *S. flexneri* infection. 293T cells expressing Flag-CAND1 were infected with different strains and immunoprecipitated CAND1 was subjected to LC-MS analyses. We observed very weak phosphorylation signals of CAND1 (likely due to endogenous kinase activities) in cells infected with the *ospG*-deletion mutant (Δ*ospG*) as well as uninfected controls (Fig. 1g). Complementation with OspG in the Δ*ospG* strain (Δ*ospG: ospG*) restored phosphorylation of CAND1, whereas introduction of a catalytic mutant (Δ*ospG: ospG* K53A) did not (Fig. 1g). Taken together, these findings suggest that CAND1 is indeed a physiological target of OspG through its kinase activity during infection.

### Phosphorylation of CAND1 disrupts its interactions with cullins

To ubiquitinate a vast array of cellular proteins, cullin-RING ubiquitin complexes (CRLs) rely on adaptor proteins (i.e., F-box proteins, FBPs) for substrate recognition[17–19]. CAND1 promotes the exchange of different FBPs in CRLs by physically interacting with cullins[21–23]. To explore potential impact of CAND1 modification, we next examined whether phosphorylation of CAND1 could interfere with its association with cullins by GST pull-down assays. We co-expressed GST-CAND1 with OspG and Ub in *E. coli* and obtained highly phosphorylated proteins (close to 100%, Supplementary Fig. 2a, b) by affinity purification and ion exchange chromatography. Upon incubation of the purified proteins with cell lysates, we then analyzed CAND1 precipitates via LC-MS for the presence of cullins. Compared to unmodified controls, pCAND1 consistently pulled down reduced amounts of cullins (including Cul1, Cul4A/B, and Cul5) (Fig. 2a and Supplementary Data 2). In line with these data, immunoblotting analyses also showed decreased association of endogenous Cul1 with pCAND1 compared to unmodified controls (Fig. 2b).

Additionally, MS analysis of highly phosphorylated CAND1 (from co-expression in *E. coli*) yielded more modification sites (S376, Y784, and Y980) besides the initially identified S558, and these residues are highly conserved in various eukaryotes (Fig. 2c, d, and Fig. S1c–f), though whether they are also phosphorylated in eukaryotic cells may

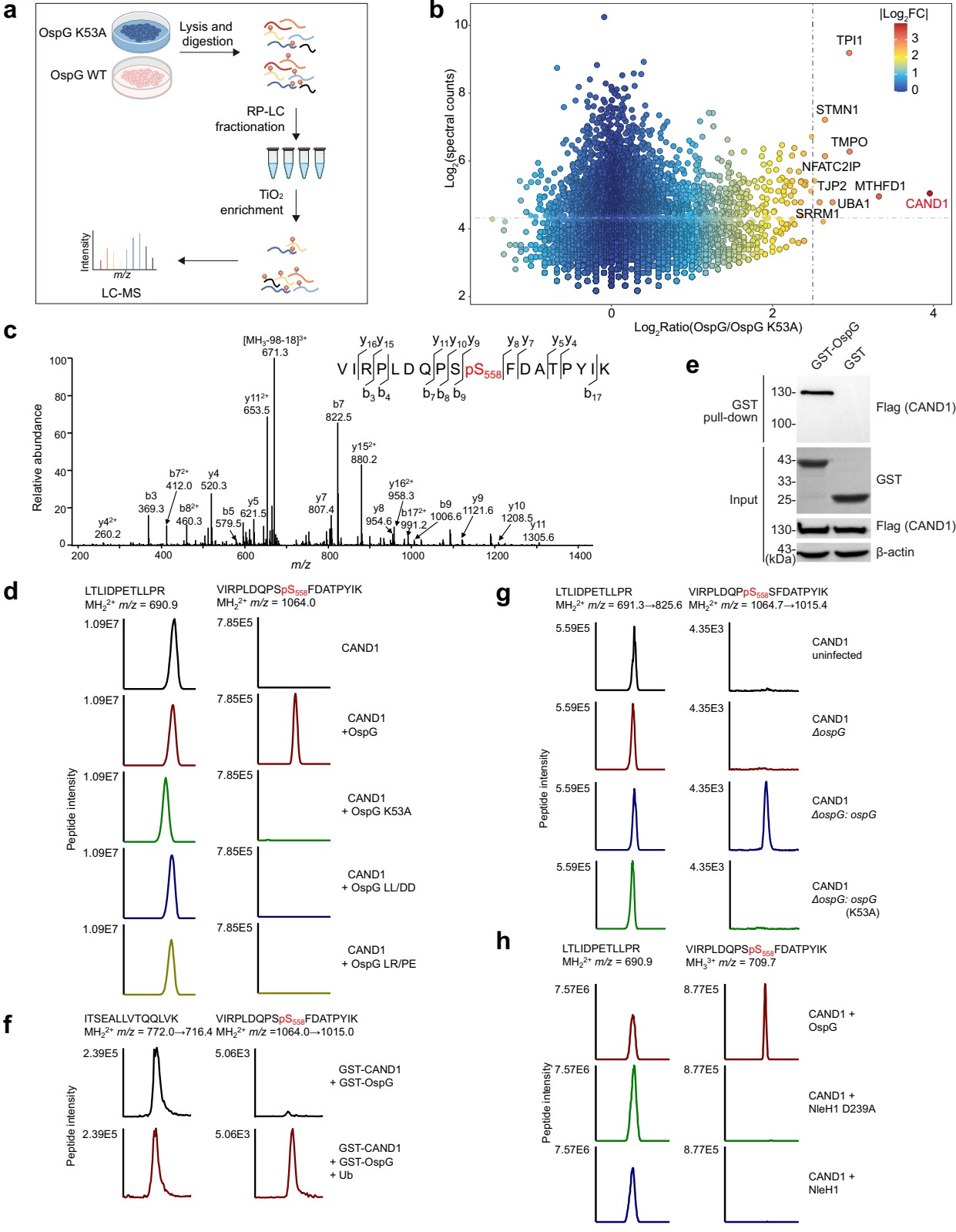

warrant further investigations. Unexpectedly, OspG catalyzed phosphorylation of both serine and tyrosine residues, suggesting this effector as a dual serine and tyrosine kinase. Next we set out to evaluate how modifications on individual residues impact CAND1-cullin associations. We constructed an array of CAND1 mutants (i.e., non-phosphorylated and phosphomimetic variants) and screened for their interactions with Cul1 by GST pull-down assays (Fig. 2e). Among those

tested mutants, the CAND1 phosphomimetic variant S376D almost lost its ability to interact with endogenous Cul1 (Fig. 2e). Consistently, co-IP experiments showed that substitution of S376 with aspartic acid completely abolished the binding of CAND1 to endogenous Cul1 (Fig. 2f). We observed similar results when the same set of samples was probed for other cullins (Supplementary Fig. 2c). Collectively, these findings support that OspG-mediated phosphorylation of CAND1

**Fig. 1 | CAND1 is a kinase substrate of OspG uncovered by large-scale host phosphoproteomics. a** A phosphoproteomic screen to identify the kinase substrates of the *S. flexneri* effector OspG. RP-LC, reversed-phase liquid chromatography. LC-MS, liquid chromatography-mass spectrometry. **b** A volcano plot depicting fold changes of detected phosphopeptides in cells expressing OspG versus OspG K53A. Candidate substrates (i.e., CAND1) of OspG appear to the far right. Proteins assigned from phosphopeptides with log₂(fold change of OspG/OspG K53A) >2.5 and summed spectral counts >20 are labeled with their names in the graph. Colors indicate the extent of fold changes (FC). **c** Collision-induced dissociation (CID)-tandem mass spectrum of CAND1 peptide bearing phosphorylation at Ser558. **d** Extracted ion chromatograms of the Ser558-phosphorylated peptide (VIRPLDQPSSFDATPYIK) and a control peptide (LTLIDPETLLPR) from indicated CAND1 samples. 293T cells were co-transfected with Flag-CAND1 and indicated OspG mutants, and immunoprecipitated CAND1 samples were resolved by SDS-PAGE before LC-MS analysis. **e** GST pull-down of Flag-CAND1 by purified GST-OspG. Glutathione resins coated with GST-OspG or GST alone were incubated

with lysates prepared from Flag-CAND1-expressing 293T cells. Eluted samples were blotted with anti-Flag and anti-GST antibodies. Images are representative of *n* = 3 independent experiments. **f** Purified GST-OspG incubated with GST-CAND1 with or without ubiquitin in vitro, and the reaction mixtures were resolved by SDS-PAGE. Potential phosphorylation signals of CAND1 were measured by SRM (selected reaction monitoring) analysis. Extracted ion chromatograms of the Ser558 phosphorylated peptide and a control peptide (ITSEALLVTQQLVK) of CAND1 were shown. **g** SRM analysis of CAND1 Ser558 phosphorylation during *S. flexneri* infection. 293T cells transfected with Flag-CAND1 were infected with indicated *S. flexneri* strains, and immunoprecipitated CAND1 samples were resolved by SDS-PAGE and analyzed by SRM. **h** Extracted ion chromatograms of the Ser558-phosphorylated peptide (VIRPLDQPSSFDATPYIK) and a control peptide (LTLIDPETLLPR) from indicated CAND1 samples. 293T cells were co-transfected with Flag-CAND1 and OspG or NleH1, and immunoprecipitated CAND1 samples were resolved by SDS-PAGE before LC-MS analysis.

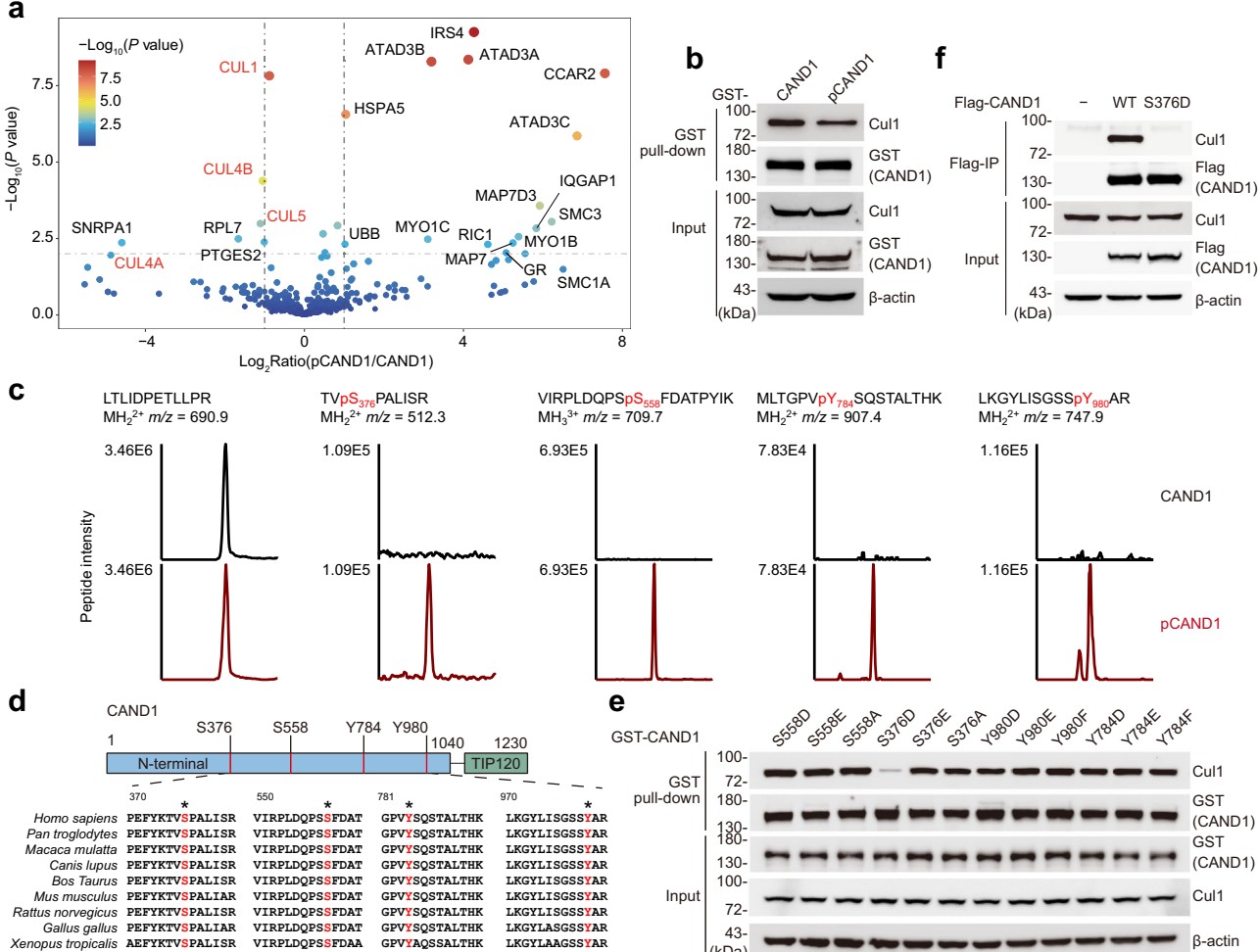

**Fig. 2 | OspG-catalyzed phosphorylation of CAND1 disrupts its interactions with cullins. a** LC-MS analysis of GST pull-down samples. The volcano plot indicates protein fold changes of pull-down samples using phosphorylated CAND1 (pCAND1) versus unmodified CAND1. Negative binomial Wald test was performed to calculate *P* values using DESeq2 package in R software, and Benjamin–Hochberg adjusted *P* values were used. Proteins with statistical significance (*P* ≤ 0.01) and fold changes of pCAND1/CAND1 >2 were labeled with names. Colors indicate *P* values. Cullins were displayed as bold red. **b** GST pull-down of 293T cell lysates by GST-tagged pCAND1 or CAND1. Eluted samples were blotted with anti-Cul1 and anti-GST antibodies to assess CAND1/pCAND1-Cul1 interactions. Images are representative of *n* = 3 independent experiments. **c** Extracted ion chromatograms of Ser376,

Ser558, Tyr784, Tyr980-phosphorylated peptides and a control peptide (LTLID-PETLLPR) from indicated GST-CAND1 samples. **d** Sequence alignment analysis of CAND1 from different species. Phosphorylated sites were highlighted in red and asterisks indicate highly conserved residues. **e** GST pull-down of 293T cell lysates by GST-tagged nonphosphorylated (Ser → Ala, Tyr → Phe) and phosphomimetic (Ser → Asp/Glu, Tyr → Asp/Glu) variants of CAND1. Eluted samples were blotted with anti-Cul1 and anti-GST antibodies. Images are representative of *n* = 3 independent experiments. **f** 293T cells were transfected with Flag-CAND1 or Flag-CAND1 S376D, and immunoprecipitated CAND1 samples were probed with an anti-Cul1 antibody to assess CAND1-Cul1 interaction. Images are representative of *n* = 3 independent experiments.

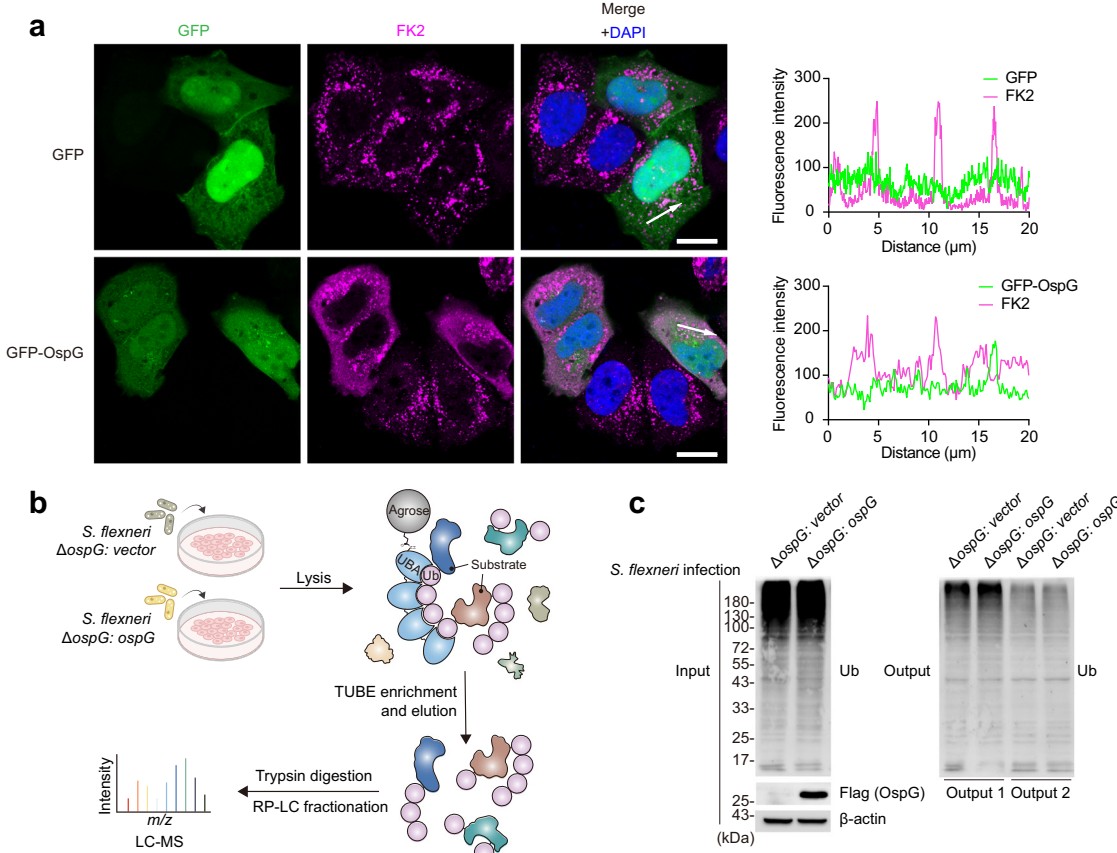

**Fig. 3 | OspG resculpts host ubiquitination landscape. a** Immunofluorescent staining of HeLa cells transfected with GFP and GFP-OspG for 24 h. The ubiquitinated proteins were stained with the FK2 antibody, and nuclei were stained with DAPI. Fluorescence intensity was plotted along the arrows. Scale bar, 20 μm. Images are representative of $n$ = 2 independent experiments. **b** Schematic diagram of the ubiquitome profiling under *S. flexneri* infection. HeLa cells were infected with indicated *S. flexneri* strains, and the NHS-activated agaroses coupled with TUBE were used to capture ubiquitinated proteins. TUBE, tandem-repeated ubiquitin-binding entities. RP-LC, reversed-phase liquid chromatography. **c** Immunoblot for ubiquitin (P4D1 antibody) showing TUBE capture of ubiquitinated proteins from cell lysates as in (**b**). Images are representative of $n$ = 3 independent experiments. Input, cell lysates before TUBE capture. Output 1 and output 2, supernatant of cell lysates after the first and the second TUBE enrichment, respectively. Ub, ubiquitin.

disrupts its association with cullins, which may ultimately translate into altered E3 ligase activities in mammalian cells.

## OspG resculpts host ubiquitination landscape
Given the role of CAND1 in swapping FBPs in CRLs, we reasoned that OspG-mediated phosphorylation of CAND1 may reprogram the CRL repertoire, leading to altered ubiquitination events. To visualize ubiquitinated cellular proteins, we stained cells using the FK2 antibody that recognizes both mono- and polyubiquitinated proteins, but not free ubiquitin. Strikingly, we observed significantly enhanced signals in OspG-transfected cells, relative to untransfected or GFP controls (Fig. 3a). To further identify those proteins that exhibit altered ubiquitin signals, we set out to globally measure ubiquitination levels of host proteins by ubiquitome profiling. Previous reports have shown the utility of tandem-repeated ubiquitin-binding entities (TUBE) for enriching ubiquitinated proteins prior to LC-MS analyses[26–28]. We adapted this strategy for profiling host ubiquitome upon *S. flexneri* infection (Fig. 3b). To screen for potential downstream targets of OspG, we contrasted the ubiquitome landscape of host cells infected by *S. flexneri* with or without OspG production. Immunoblotting analyses showed that tandem TUBE pull-down efficiently enriched most ubiquitinated proteins (Fig. 3c).

In total, we detected >2800 ubiquitinated protein candidates, of which ~200 assignments were statistically different (Supplementary Data 3). More than 50 proteins exhibited higher levels (>1.5-fold) in

cells infected by OspG-expressing bacteria but not the Δ*ospG* strain. In addition, more than half of the altered proteins had lower ubiquitination levels as well (Fig. 4a). Gene ontology (GO) analysis revealed the enrichment of diverse biological processes and cellular functions including the NF-κB pathway (e.g., TRAF2, IKBKG) (Fig. 4b). Further KEGG pathway analysis highlighted proteins in pathogen infection, protein processing and metabolism in addition to NF-κB signaling (Fig. 4c). Overall, these data support the notion that phosphorylation of CAND1 by OspG likely results in a global perturbation of substrate ubiquitination via CRLs.

## Septin family proteins undergo increased ubiquitination with mixed types of Ub chains
Notably, our data revealed that a family of cytoskeletal proteins known as septins (including SEPT2, SEPT7, SEPT9, SEPT10 and SEPT11) consistently exhibited higher levels of ubiquitination in cells infected by OspG-expressing *S. flexneri* relative to those controls (Fig. 4a). To verify these findings, we co-expressed individual Flag-tagged septins in 293T cells together with GFP, OspG, or OspG K53A. Then immunoprecipitated septin samples were probed for their ubiquitination levels with an α-Ub antibody. Given the formation of SEPT2-SEPT6-SEPT7-SEPT9 hetero-octamer that assembles into filaments[29,30], we included SEPT6 in our assays as well. Other than SEPT2, SEPT7 and SEPT11 with minimal signals, SEPT6, SEPT9, and SEPT10 underwent robust ubiquitination when co-expressed with OspG but not GFP controls, while

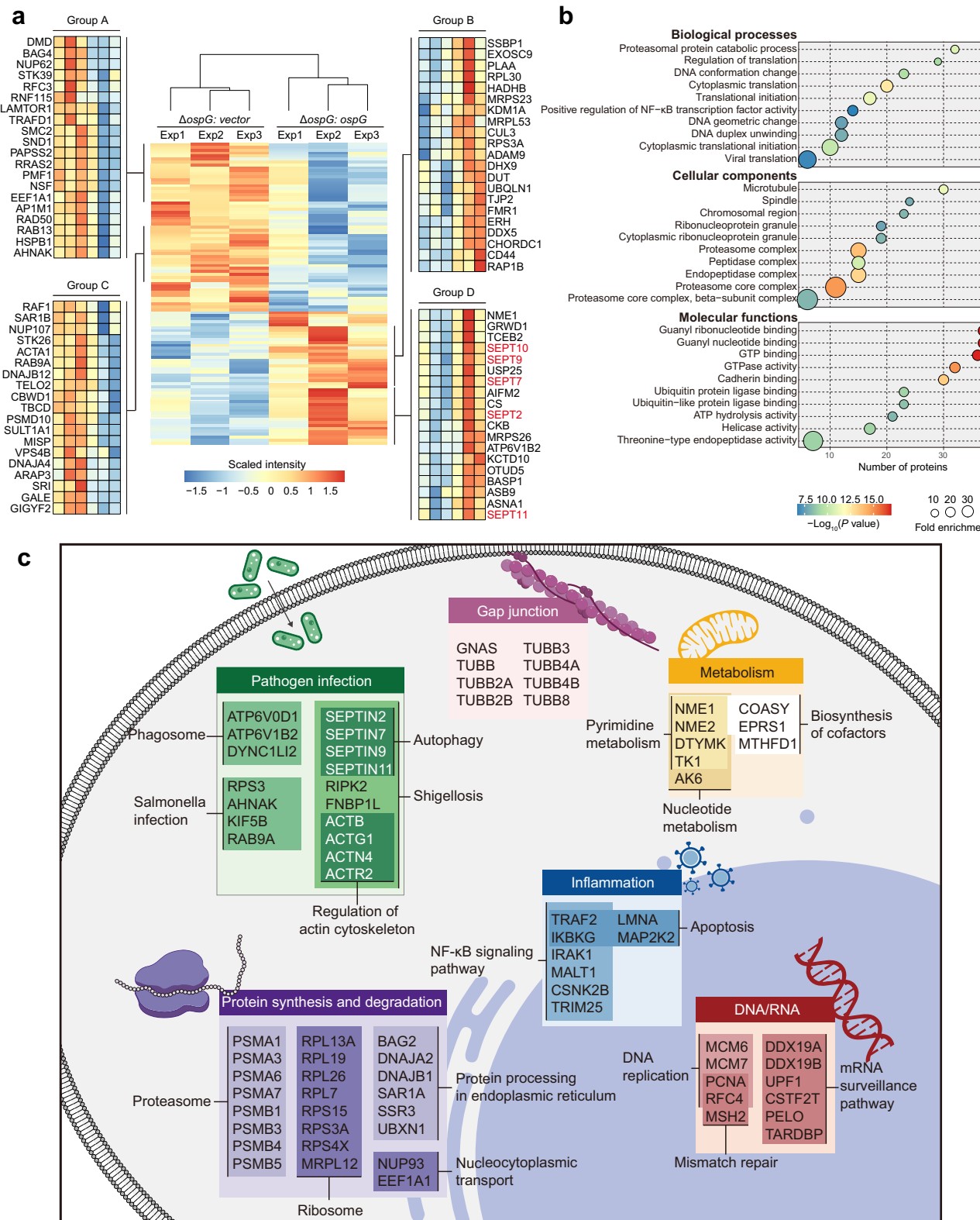

**Fig. 4 | The ubiquitome profiling of *S. flexneri* infected cells. a** Heatmap illustrating statistically significant changes in the ubiquitome of *S. flexneri* infected cells (Δ*ospG: ospG* versus Δ*ospG: vector*). Colors indicate scaled intensity of LC-MS identified proteins. **b** Dot plot of Gene Ontology (GO) enrichment showing significantly over-represented GO terms for statistically significant ($P \leq 0.05$) changes in the ubiquitome of *S. flexneri* infected cells (Δ*ospG: ospG* versus Δ*ospG: vector*). $P$ values were calculated by using one-sided hypergeometric test implemented in R package clusterProfiler without adjustments. Colors indicate $P$ values, and dot sizes indicate fold enrichment compared to the whole proteome. Dots were not shown for terms without statistical significance ($P > 0.05$). **c** DAVID functional clustering of the statistically significant ($P \leq 0.05$) changes of *S. flexneri* infection ubiquitome was performed to identify Kyoto Encyclopedia of Genes and Genomes (KEGG) pathways.

intermediate levels were observed in cells producing the catalytic mutant (Fig. 5a). Likewise, moderate ubiquitination levels were detected for SEPT9 co-expressed with the LL/DD, and LR/PE mutants incapable of interacting with Ub or E2, likely due to their inability to promote the kinase activity of OspG and hence CAND1 phosphorylation (Fig. 5c). SEPT9 ubiquitination were further attenuated with minimal signals in cells expressing the double mutants (LL/DD K53A or LR/PE K53A) (Fig. 5c).

Consistent with the above findings that OspG promotes septin ubiquitination, we also detected increased signals of polyubiquitin-conjugated SEPT9 in OspG-expressing cells relative to GFP controls when TUBE pull-down samples were further probed with an antibody specific for SEPT9 (Supplementary Fig. 3a). In line with previous immunoblotting data (Fig. 5a), we didn't observe any visible signals for polyubiquitinated SEPT7, though it was more abundant in TUBE pull-down samples from OspG-producing cells, likely due to its association with other ubiquitinated proteins (Supplementary Fig. 3a). Taken together, these data indicate that OspG promotes ubiquitination of host septins, at least a subset of them, which is partially dependent on its kinase activity.

Next we set out to investigate the potential consequence(s) of OspG-dependent ubiquitination on septin biology. As SEPT9 has robust ubiquitination and complex domain organization, we mapped its modification sites by LC-MS and found multiple ubiquitinated residues including K62, K69 and K579 (Supplementary Fig. 3b–d). These modified sites are located in both its N-terminal and C-terminal domains responsible for the association with actin filaments and microtubules (Supplementary Fig. 3e). Simultaneous mutation of these three lysines did not noticeably reduce the level SEPT9 ubiquitination, suggesting the presence of other unidentified modification sites (Supplementary Fig. 3f). Immunoblotting data showed that septins underwent polyubiquitination (poly-Ub) (Fig. 5a). Different Ub chains can attach to either one of its seven lysines (K) or the N-terminal free amino group. Of note, K48- and K63-linked chains are most characterized and K48 Ub linkage often signals for proteasomal degradation. Next we sought to determine the type of poly-Ub chain linkage on SEPT9. We co-expressed OspG, SEPT9 and a Ub variant from two sets of mutants. In the first set, one of the seven lysines was substituted with arginine. Replacement of K6, K27, and K33 with arginine substantially reduced the ubiquitination signals of SEPT9 (Fig. 5b). With the other set of variants, the substitution of all lysines but K6 yielded comparable ubiquitination levels. Similar findings were observed when K27 or K33 was left out as the only lysine available for Ub conjugation (Fig. 5b). Taken together, these data suggest that ubiquitinated SEPT9 likely contains mixed K6-, K27- and K33-linked Ub chains, consistent with a non-canonical role of ubiquitination (i.e., other than protein degradation). Consistently, we found that the abundance of endogenous SEPT9 in cells expressing OspG or OspG K53A was not altered relative to GFP controls (Supplementary Fig. 4a). Similar findings were observed in infected host cells as well (Supplementary Fig. 4b).

Furthermore, we tested if SEPT9 ubiquitination would affect its interaction with other septin members. We expressed individual Flag-tagged septins (2, 6 and 7) in 293T cells either alone or with GFP-OspG. Immunoprecipitation with anti-Flag resins revealed reduced association of SEPT9 with SEPT6 and SEPT7, but not SEPT2 when OspG was produced (Fig. 5d). These data suggest that OspG-mediated poly-ubiquitination likely disrupts the interactions of SEPT6/7 with SEPT9, consequently impairing their oligomerization.

### Ubiquitination of septins prevents the assembly of septin cages during infection

Like actin or tubulins, septins are integral components of the cytoskeleton that assembles into filament structures[31,32]. Previous reports have shown that septin assemblies are recruited to cytosolic bacteria that polymerize actin such as *S. flexneri*[33]. Notably, septin cage-like structures entrap cytosolic bacteria, thereby limiting pathogen dissemination via cell-to-cell spread and targeting to autophagy[34–36]. These findings inspired us to explore whether septin ubiquitination would regulate the assembly of septin cages. In uninfected cells, immunostaining against endogenous SEPT9 features typical filamentous structures with lattice-like arrangement around cell periphery. Consistent with the previous reports[33], we observed the formation of compact septin aggregates around individual *S. flexneri*, accounting for approximately 15% of cytosolic bacteria (Fig. 6a–b). Strikingly, we found substantially more cage-like structures (25%) in cells infected by the *ospG*-deletion mutant (Δ*ospG*). In addition, cells infected by the Δ*mxiE* strain exhibited septin-cages in a level comparable to that in WT infection (Fig. 6b), in agreement with the previous study[37]. Though MxiE transcriptionally regulates the expression of many effector proteins[38], proteomic analysis of relevant strains revealed that OspG is likely expressed in a MxiE-independent manner (Supplementary Fig. 5b). Similar findings were observed in cells when cage-like structures were visualized by SEPT7 staining (Fig. 6b, Supplementary Fig. 5a). Together, these data suggest a prominent role of OspG in preventing septin-cage assembly. Additionally, the re-introduction of OspG into the Δ*ospG* strain restored the capability of blocking septin-cage assembly (Fig. 6c–d). When complemented with the catalytic mutant (K53A), we observed septin-cages at an intermediate level, suggesting a role of kinase activity in this process (Fig. 6d).

To further support a role of septin ubiquitination in antagonizing cage assembly, we stained both septin cages and polyubiquitin chains. We found co-localization of the FK2 signals with SEPT9 in cages that entrapped WT but not Δ*ospG* bacteria (Supplementary Fig. 6a, b). Additionally, these cages with overlapping FK2 and SEPT9 signals seemed to be deformed, likely representing cages undergoing ubiquitination-mediated disintegration. In contrast, such co-localization was barely observed in intact septin cages. Collectively, these findings suggest that *S. flexneri* has evolved a mechanism of blocking septin-cage assembly via ubiquitination in an OspG-dependent manner (Fig. 7).

## Discussion

Ubiquitin signaling regulates most biological processes in eukaryotic cells. Here we report that a bacterial kinase OspG co-opts host ubiquitination pathways to inhibit the formation of septin-cages and escape host entrapment. Mechanistically, the *S. flexneri* type III effector OspG, by interacting with E2 enzymes and Ub, recruits host ubiquitination machinery comprising cullin-RING ubiquitin ligases (CRLs) and CAND1 to ubiquitinate septin family proteins. Furthermore, OspG phosphorylates CAND1 through its kinase activity that is stimulated by interacting with E2-Ub. Phosphorylated CAND1 exhibits reduced association with cullins, fine-tuning the extent of substrate ubiquitination such as septins. Finally, OspG-mediated septin modifications confer *S. flexneri* the ability to block the assembly of septin-cages in host cells (Fig. 7).

Most proteins undergo ubiquitination at least one time in their life time. To ensure precise targeting of such a vast array of substrates, cells elaborate a large repertoire of dynamic ubiquitin E3 ligases. Among these, cullin-RING ubiquitin ligases (CRLs) represent the largest class of E3 enzymes, of which the SCF complex is best studied[16]. The modular SCF ubiquitin ligases feature a large number of FBPs recognizing diverse targets[17–19]. CAND1 promotes the exchange of various FBPs to dynamically regulate the cellular CRL repertoire[20–23]. Although prokaryotes do not code for the canonical ubiquitin system, many pathogenic bacteria co-opt the host ubiquitin network by virulence factors harboring E3 ubiquitin ligase or deubiquitinase (DUB) activities or activities that directly attack components of the ubiquitination machinery, including Ub itself[39–41].

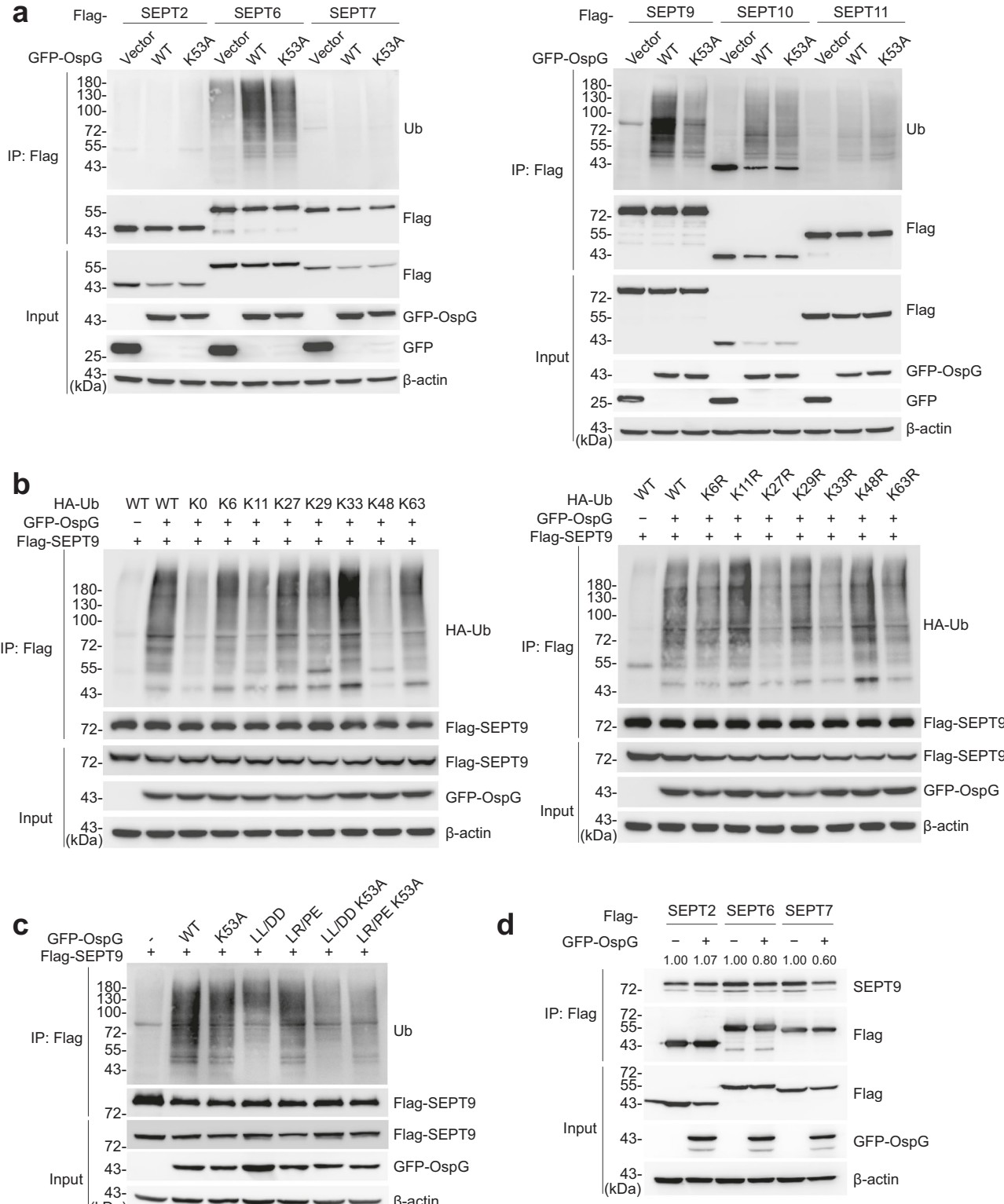

**Fig. 5 | The septin family proteins exhibit elevated ubiquitination levels in an OspG-dependent manner. a** Ubiquitination of septin-family members in 293T cells co-expressing GFP, GFP-OspG or GFP-OspG K53A. Lysates were immunoprecipitated with Flag antibody and immunoblotted as indicated. Images are representative of *n* = 5 independent experiments. **b** Ubiquitination of SEPT9 in Flag-SEPT9-expressing cells co-transfected with GFP-OspG and indicated HA-tagged ubiquitin constructs. Images are representative of *n* = 2 independent experiments. The GFP vector was used as a control. K0 means ubiquitin in which all lysines were mutated to arginines and K6 means ubiquitin in which all lysines except Lys6 were mutated to arginines. K6R means ubiquitin in which only Lys6 were mutated to arginine. **c** Ubiquitination of SEPT9 in Flag-SEPT9-expressing cells co-transfected with GFP-OspG and indicated mutants. Images are representative of *n* = 3 independent experiments. **d** 293T cells transfected with individual septins (SEPT2, SEPT6, and SEPT7) either alone or with GFP-OspG. Lysates were immunoprecipitated with Flag antibody and further immunoblotted as indicated. Images are representative of *n* = 2 independent experiments.

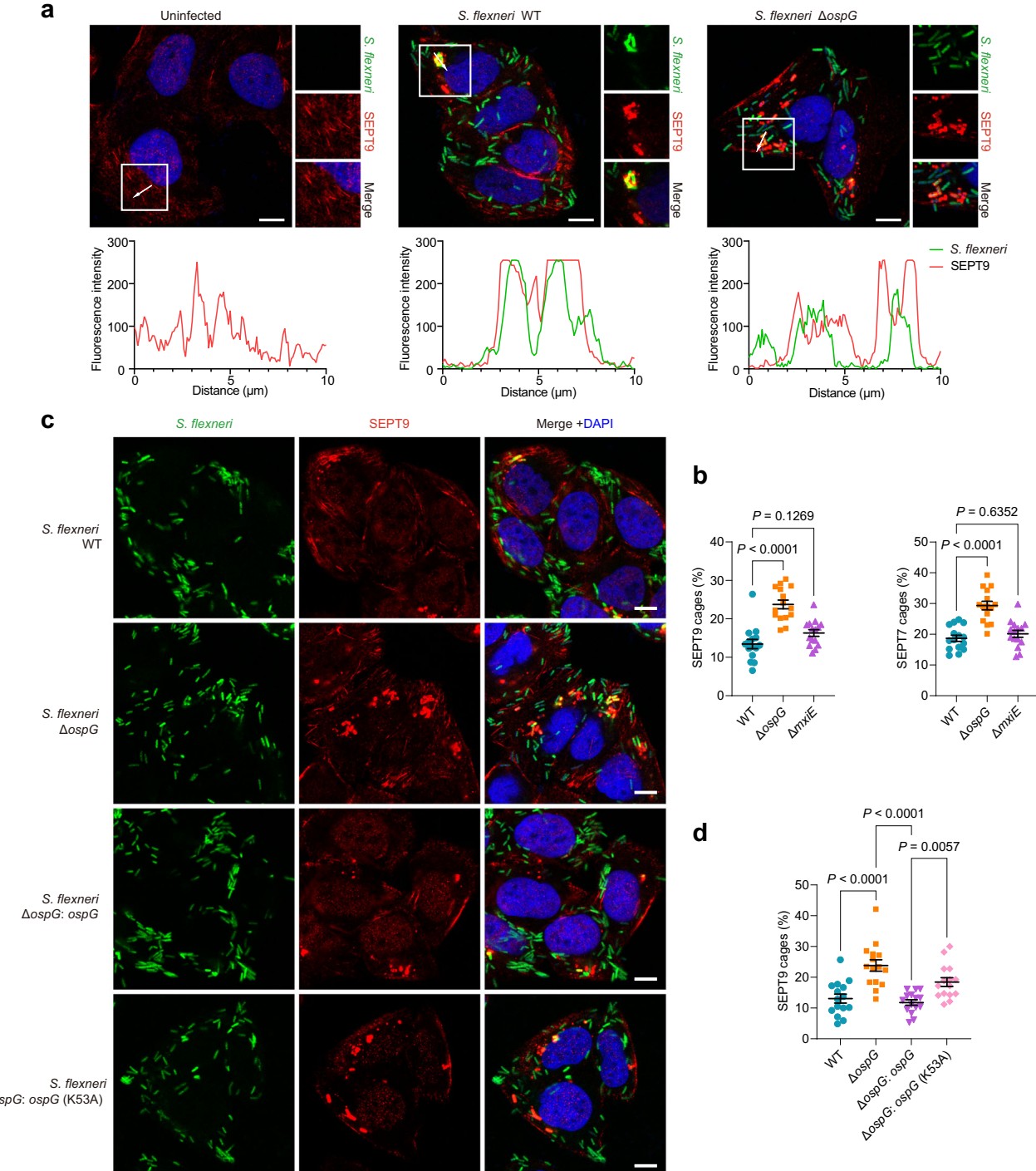

**Fig. 6 | OspG-promoted ubiquitination of septins prevents the assembly of septin-cages around cytosolic *S. flexneri* in infected host cells. a** Representative immunofluorescence images of uninfected or *S. flexneri*-infected cells. HeLa cells were infected with indicated GFP-labeled *S. flexneri* strains for 2 h, fixed and stained with an anti-SEPT9 antibody. Nuclei were stained with DAPI. Fluorescence intensity was plotted along the arrows. Scale bars, 10 μm. **b** HeLa cells were infected with *S. flexneri* at an MOI of 10 for 2 h, fixed and stained with anti-SEPT9 or anti-SEPT7 antibodies for quantitative microscopy. Data represent the mean % ± SEM of *S. flexneri* inside septin-cages from *n* = 3 biologically independent experiments and a total of 150 HeLa cells in which the proportion of septin-cages was counted. One-

way ANOVA followed by Dunnett's multiple comparison test was performed. **c** Representative immunofluorescence images of HeLa cells infected with indicated GFP-labeled *S. flexneri* (MOI = 10) strains for 2 h, fixed and stained with an anti-SEPT9 antibody. Nuclei were stained with DAPI. Scale bars, 10 μm. **d** HeLa cells were infected with *S. flexneri* at an MOI of 10 for 2 h, fixed and stained with an anti-SEPT9 antibody for quantitative microscopy. Data represent the mean % ± SEM of *S. flexneri* inside septin-cages from *n* = 3 biologically independent experiments and a total of 150 HeLa cells in which the proportion of septin-cages was counted. One-way ANOVA followed by Dunnett's multiple comparison test was performed.

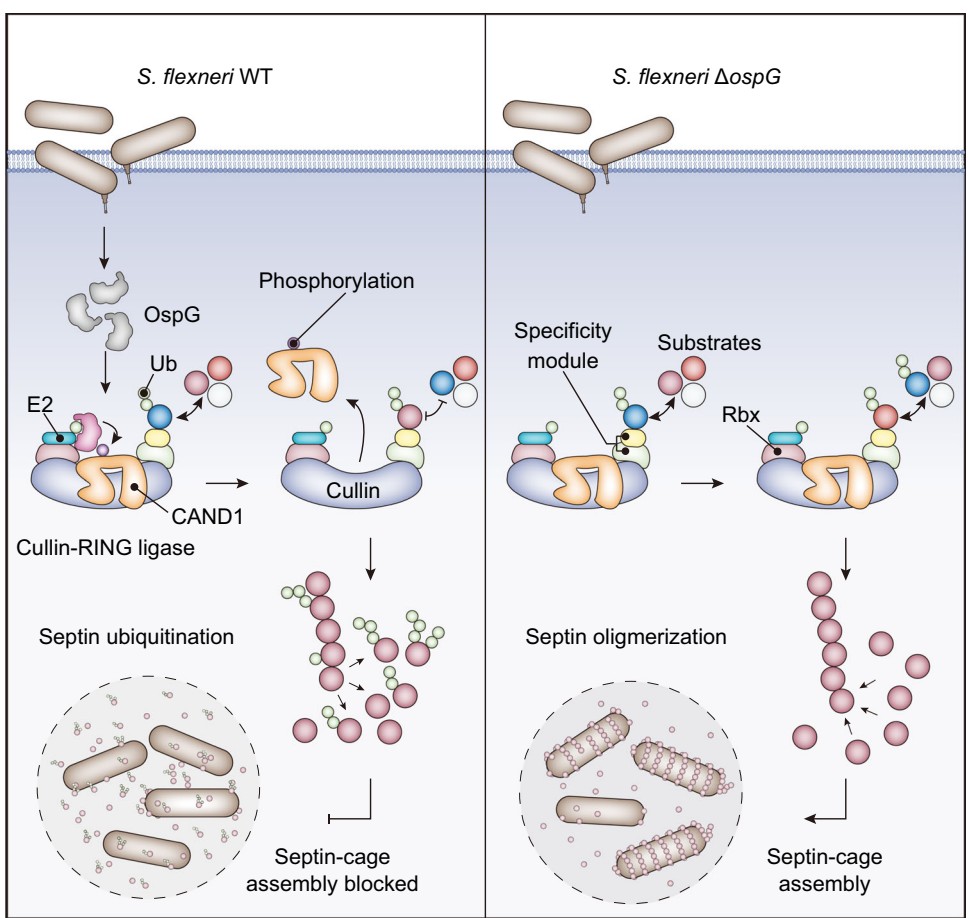

**Fig. 7 | A proposed model illustrating that OspG-mediated phosphorylation of CAND1 disrupts the balance of CRL-mediated ubiquitination of protein substrates, thereby promoting septin ubiquitination and preventing the assembly of septin-cages around bacterial pathogens.**

Our study found that a bacterial kinase OspG modulates SCF dynamics by targeting CAND1 via phosphorylation in a process that requires its association with E2 and Ub. Among multiple modified serine and tyrosine residues, CAND1 S558 phosphorylation was also reported by a previous phosphoproteomics survey, raising the possibility of a yet unidentified endogenous kinase as well as phosphorylation-dependent regulation of CAND1/SCF functions in mammalian cells[42].

We further provide evidence that CAND1 phosphorylation disrupts its interaction with cullins, the scaffold protein in SCF, leading to altered SCF landscape and ubiquitination events. Consistent with this notion, our ubiquitome data uncovered ubiquitinated substrates that were modified in an OspG-dependent manner. We demonstrated that OspG promotes poly-ubiquitination of septin family proteins with mixed types of noncanonical Ub chains, though partially dependent on its kinase activity. Interestingly, previous studies have shown elevated phosphorylation of SEPT9 in cells infected by *S. flexneri* and EPEC[43,44]. Indeed we were able to detect robust signals of phosphorylated SEPT9, though its levels seemed to be independent of OspG (Supplementary Fig. 4c, d). Functionally, OspG-mediated ubiquitination antagonizes a unique host defense strategy of entrapping cytosolic pathogens via septin cage-like structures.

Overall, our current study provides novel insights into the delicate regulation of host ubiquitin signaling and septin biology by *S. flexneri* via a kinase effector, further expanding the spectrum of cellular processes regulated by ubiquitination. Importantly, fine-tuning the activity of SCF/CRL complexes through CAND1 modifications (i.e., phosphorylation and ubiquitination) might be a widespread regulatory mechanism in eukaryotes. Further studies are warranted to explore additional layers of regulation on CRLs.

## Methods

### Plasmids and antibodies

DNA for *ospG* was amplified from *S. flexneri* 2a strain 2457T. Complementary DNAs (cDNA) for human CAND1, SEPT2, SEPT6, SEPT7, SEPT9 (isoform a), SEPT10, and SEPT11 were amplified from reverse-transcribed cDNA of 293T cells. cDNA for TUBE was amplified from pGEX-6P-1-TUBE, a construct with 4×UBA domains from UBQLN1, a kind gift from Dr. Meiping Zhao's laboratory (Peking University). A set of constructs with Ub mutants were kindly provided by Dr. Jianyuan Luo (Peking University). The DNAs were cloned into pcDNA3.1-Flag, pcDNA4-HA, pcDNA4-Flag-HA, pCS2-3×Flag, pCS2-3×HA or pCS2-GFP vectors for transient expression in mammalian cells. For recombinant protein expression in *E. coli*, DNAs for OspG, Ub, CAND1 and their mutants were cloned into pGEX-6P-2, pET21a or pET28a vector. For complementation in *S. flexneri* Δ*ospG* strain, C-terminal Flag-tagged DNAs for *ospG* and its mutants were cloned into pME6032 or pBBR1MCS2 vector. All mutations were generated by the standard polymerase chain reaction (PCR) and all plasmids were verified by sequencing.

The following primary antibodies were used in this study: anti-Cul1 (WB 1:200, Santa Cruz, sc-12761), anti-Cul4A (WB 1:5000, abcam, ab92554), anti-Cul4B (WB 1:1000, Abclonal, A12696), anti-Cul5 (WB 1:5000, abcam, ab184177), anti-SEPT7 (IF 1:100, WB 1:2000, Proteintech, 13818-1-AP), anti-SEPT9 (IF 1:300, WB 1:3000, Atlas Antibodies, HPA042564), anti-ubiquitinated proteins, clone FK2 (IF 1:400,

Sigma, 04-263), anti-Ubiquitin (WB 1:200, P4D1) (Santa Cruz, sc-8017), anti-CAND1 (WB 1:5000, abcam, ab183748), anti-Flag (WB 1:3000, Zen Bioscience, R24091), anti-HA (WB 1:5000, Invitrogen, 26183), anti-GST (WB 1:2500, Zen Bioscience, 390028), anti-GFP (WB 1:5000, Easybio, BE2001), anti-β-actin (WB 1:2000, Cell Signaling Technology, 4967). The secondary antibodies horseradish peroxidase (HRP)-conjugated anti-mouse IgG (WB 1:5000,115-035-003), HRP-conjugated anti-rabbit IgG (WB 1:5000, 111-035-003) and Cyanine3 (Cy3)-conjugated anti-rabbit IgG (IF 1:500, 111-165-003) were purchased from Jackson ImmunoResearch Laboratories. Alexa Fluor 647-labeled anti-mouse IgG (IF 1:500, A0473) were purchased from Beyotime biotechnology.

### Bacterial strains

*S. flexneri* 2a strain 2457T was kindly provided by Dr. Feng Shao (National Institute of Biological Sciences, China). *S. flexneri* was freshly streaked from glycerol stocks onto tryptone soy agar supplemented with 0.02% (w/v) congon red, and a single red colony was selected and inoculated into tryptone soy broth with appropriate antibiotics. For efficient infection of host cells, all *S. flexneri* strains used in this study were transformed with a plasmid expressing afimbrial adhesin (Afa).

*S. flexneri* Δ*ospG* strain were generated by standard homologous recombination using the suicide vector pSR47s. The upstream and downstream 800 bp DNA sequences of the deleted fragment were amplified, assembled through overlap-PCR and inserted into pSR47s to generate pSR47s Δ*ospG* plasmid. After confirming the inserted sequence by DNA sequencing, the plasmid was transferred to *S. flexneri* via conjugation using *E. coli* SM10λpir strain on Luria-Bertani (LB) broth without antibiotics. After conjugation, the transconjugants were screened on LB agar plates supplemented with 50 μg/mL kanamycin and 30 μg/mL streptomycin. The integrants were further screened by LB agar plates containing 15% sucrose. The final mutant strains were confirmed by PCR using two different pairs of primers and further DNA sequencing of the PCR products.

*S. flexneri* Δ*mxiE* strain were generated via a CRISPR-Cas9 system[45]. Briefly, cDNA of gRNA (5′-AAGAGTCTGACAGAGCATTA-3′) for *mxiE* was designed by the CRISPOR website and was inserted into the pTarget plasmid. The upstream and downstream 500 bp DNA sequences of the fragment upon *mxiE* deletion (90-543 bp of *mxiE* were deleted) were amplified, assembled through overlap-PCR. The pCas plasmid containing λ-RED and Cas9 gene was transformed into *S. flexneri* 2a strain 2457T, and positive colonies were selected to induce the expression of λ-RED and make competent cells. Next, the pTarget plasmid and homologous fragment of Δ*mxiE* were electroporated into the competent cells. The individual colonies were verified by colony PCR and DNA sequencing. Finally, the verified colonies were used for pTarget and pCas plasmids curing by adding 0.5 mM IPTG and grown at 42 °C, respectively.

*E. coli* strains were grown in LB broth at standard temperatures (37 °C for growth, 18 °C, or 22 °C for protein expression induction) with appropriate antibiotics.

### Cell lines, transfection, and immunoprecipitation

293T (Cat No. CRL-3216) and HeLa cells (Cat No. CRM-CCL-2) were obtained from American Type Culture Collection (ATCC) and cultured in Dulbecco's modified Eagle's medium (DMEM) supplemented with 10% fetal bovine serum (TRANS, FS301-02) at 37 °C in a 5% CO$_2$ incubator. Cells were checked for potential mycoplasma infection and tested negative. Transient transfection was performed using Hieff Trans liposomal transfection reagents (YEASEN biotech, 40802ES03) or polyethyleneimine (PolySciences) following the manufacturers' instructions.

For immunoprecipitation, cells were harvested and lysed in the lysis buffer (50 mM Tris-HCl, pH 7.5, 150 mM NaCl, 1% Triton X-100). Cell lysates were cleared by centrifugation at 13,000 × *g*, 4 °C for 10 min twice, the supernatants were incubated with pre-washed anti-

Flag M2 affinity beads (Sigma-Aldrich, A2220) with gentle rotation for 4 h at 4 °C. The beads were washed three times with lysis buffer, and proteins were eluted with Flag or 3×Flag peptides.

### Cell culture infection

For enriching the ubiquitinated proteome by TUBE pull-down, HeLa cells were seeded in 10-cm dishes 12–16 h before infection. *S. flexneri* (with Afa) was grown overnight at 37 °C in tryptone soy broth with shaking. Bacterial cultures were diluted 1:20 in fresh tryptone soy broth and grown at 37 °C until OD$_{600}$ reached 1.0. Bacteria were pelleted at 3000 × *g* and re-suspended in Hank's Balanced Salt Solution (HBSS), and were added to cells at a multiplicity of infection (MOI) of 100. Infection was proceeded for 1 h at 37 °C in a 5% CO$_2$ incubator, cells were then washed twice with HBSS, once with DMEM containing 100 μg/mL gentamicin, and overlaid with DMEM containing 100 μg/mL gentamicin to kill the extracellular bacteria. Two hours later, cells were harvested for the subsequent TUBE pull-down.

For microscopy analyses of septin-cage, HeLa cells were seeded onto glass coverslips in 24-well plates and cultured for 12–16 h before infection. Overnight bacterial cultures were diluted 1:50 in fresh tryptone soy broth and grown at 37 °C until OD$_{600}$ reached 0.6, and cell monolayers were infected with *S. flexneri* (with Afa) at an MOI of 10 for 30 min. Infected cells were washed 3 times with DMEM and then switched into fresh DMEM containing 50 μg/mL gentamicin. Cells were fixed with 4% paraformaldehyde at the indicated time points after bacterial infection.

For detection of CAND1 phosphorylation during bacterial infection, 293T cells transfected with 3×Flag-CAND1 in 6-well plates for 24 h were infected with *S. flexneri* (with Afa) at an MOI of 100 for 1 h, and then switched into fresh DMEM containing 100 μg/mL gentamicin. Two hours later, cells were harvested and were lysed in lysis buffer containing 50 mM Tris-HCl (pH 7.5), 150 mM NaCl, 1% Triton X-100 supplemented with phosphatase inhibitor cocktail. The lysates were incubated with anti-Flag M2 resins for 4 h at 4 °C. The resins were washed three times with lysis buffer, and the samples were run on SDS-PAGE and processed further for subsequent LC-MS analysis.

### Phosphoproteomics

293T cells transfected with plasmids expressing OspG or OspG K53A for 24 h were harvested and lysed in the lysis buffer (8 M urea, 50 mM ammonium bicarbonate) supplemented with phosphatase inhibitor cocktail. Cells lysis were briefly sonicated and centrifuging at 13,000 × *g* for 30 min at 4 °C. Protein concentration was determined by BCA protein assay. A total of 3 mg of proteins were incubated with 5 mM dithiothreitol at 56 °C for 30 min, and alkylated with 25 mM iodoacetamide at room temperature for 1 h in darkness. The samples were diluted with 50 mM ammonium bicarbonate to a final urea concentration of 2 M, and digested with trypsin (Sigma) overnight at an enzyme-to-protein ratio of 1:50 (w/w) at 37 °C. Finally, the digestion was terminated and acidified by the addition of TFA (1%). The peptide samples were desalted on C18 SepPak solid-phase extraction cartridges (Welch Materials, 00559-11002) and vaccum dried. Reconstituted peptides were further fractionated into seven fractions by reversed-phase liquid chromatography (InertSustain C18 column, 2.1 mm × 50 mm, 5 μm particle size, SHIMADZU) under basic pH on an offline HPLC instrument (SHIMADZU, LC-20AT). The fractions were further dried prior to phosphopeptide enrichment.

Phosphopeptides were enriched with 5 μm titanium dioxide (TiO$_2$) beads (GL Sciences, Japan) that were resuspended with a binding buffer containing 65% acetonitrile (ACN), 2% trifluoroacetic acid (TFA), and saturated glutamic acid and incubated on a shaker at 4 °C for 10 min before use. Peptides were incubated with TiO$_2$ beads at a peptide-to-bead ratio of 1:3 (w/w) for 30 min. Afterward the beads were washed with 80% ACN and 0.5% TFA. Phosphopeptides were eluted with 10% ammonia solution prior to LC-MS analysis.

## Protein expression and purification

The *E. coli* BL21 (DE3) strains transformed with expression plasmids were grown in LB broth supplemented with kanamycin (50 µg/mL) or ampicillin (100 µg/mL). Protein expression was induced by the addition of 0.4 mM isopropyl β-d-1-thiogalactopyranoside (IPTG) when the $OD_{600}$ of cell culture reached approximately 0.8, and the cells were grown at 22 °C (18 °C for expression of CAND1 and its variants) for 16 h. For purification of GST-OspG and its variants, GST-CAND1 and its variants, bacterial cells were harvested and lysed by an ultrasonic homogenizer in buffer containing 50 mM Tris-HCl (pH 7.5), 150 mM NaCl, 2 mM dithiothreitol. Proteins were purified using glutathione resins (Sangon Biotech, C600031). For purification of 6×His-tagged proteins, bacterial cells were harvested and lysed by an ultrasonic homogenizer in buffer containing 20 mM Tris-HCl (pH 8.0), 300 mM NaCl, 20 mM imidazole, 2 mM dithiothreitol. Proteins were purified using HisTrap HP affinity chromatography (GE Healthcare Life Sciences). Further purification was carried out via Superdex 75 10/300 GL gel filtration chromatography and HiTrap Q HP anion exchange chromatography (GE Healthcare Life Sciences) if needed. The purified proteins underwent buffer exchange via HiTrap Desalting chromatography (GE Healthcare Life Sciences) before concentration and storage at −80 °C in a buffer containing 50 mM Tris-HCl (pH 7.5), 150 mM NaCl, 2 mM dithiothreitol, and 10% glycerol.

## In vitro phosphorylation reactions

Purified OspG and CAND1 proteins (molar ratio 1: 1) were added to the reaction mixture with or without the addition of the same molar ubiquitin. The reaction was carried out at 30 °C for 30 min in a buffer containing 25 mM HEPES (pH 7.5), 150 mM NaCl, 5 mM $MgCl_2$, 1 mM dithiothreitol, 1 mM ATP. SDS sample loading buffer was added to the reaction mixture to stop reaction, and samples were loaded onto SDS-PAGE gel for Coomassie blue staining and subsequent LC-MS analysis.

## GST-CAND1 pull-down assays

Recombinant GST-tagged nonphosphorylated and phosphorylated CAND1 were expressed by co-transforming *E. coli* BL21 (DE3) strain with pCOLADuet-1-OspG WT/K53A-Ub and pGEX-6P-2-CAND1. Concentrations of purified recombinant proteins were determined by Bradford assays and the modification rate of GST-tagged phosphorylated CAND1 was analyzed by LC-MS. For pull-down assays, 293T cells were grown to 90–100% confluence and were lysed on ice in lysis buffer containing 50 mM Tris-HCl (pH 7.5), 150 mM NaCl, 1% Triton X-100 supplemented with a cocktail of phosphatase inhibitors. Cell lysates were centrifuged twice at 13,000 × *g* for 10 min, and the supernatants were subjected to Bradford assays to determine total protein concentration. Lysates were first incubated with empty glutathione resins at 4 °C for 30 min to minimize non-specific binding, which were then divided equally into desired aliquots. Each aliquot was incubated with approximately 1 µg of GST-tagged CAND1 or its variants per mg of lysates at 4 °C for 10 min respectively, and the mixtures were then incubated with 50 µL of glutathione resins at 4 °C for 1 h. The beads were washed three times with ice-cold lysis buffer, and bound proteins were eluted by boiling in 2×SDS-PAGE loading buffer. Western blot analysis was performed immediately after elution, and the remaining samples were separated by SDS-PAGE for Coomassie blue staining and processed for subsequent LC-MS analysis.

## TUBE protein conjugation and TUBE pull-down

Purified 6×His-tagged TUBE protein was first applied to HiTrap Desalting chromatography for buffer exchange to coupling buffer containing 0.2 M $NaHCO_3$, 0.5 M NaCl (pH 8.0) before conjugated to NHS-activated Sefinose (Sangon Biotech, C600024) following the manufacturer's instructions. The TUBE-conjugated beads were stored at 4 °C in PBS containing 20% ethanol.

For TUBE-based ubiquitome analysis, cells from four 10-cm dishes of HeLa cells after *S. flexneri* infection per sample were lysed in 4 mL of lysis buffer containing 50 mM Tris-HCl (pH 7.5), 150 mM NaCl, 1% Triton X-100, 1 mM ethylenediaminetetraacetic acid (EDTA), 1 mM iodoacetamide for 15 min at 4 °C. The lysates were centrifuged at 13,000 × *g* for 10 min at 4 °C. The supernatant was incubated with 80 µL (packed volume) of TUBE-conjugated beads per sample for 30 min at 4 °C. The mixture was centrifuged at 1000 × *g* for 1 min and the supernatant was incubated with another 80 µL (packed volume) of TUBE-conjugated beads for 30 min at 4 °C. Beads with captured proteins were then washed three times with 1 mL of lysis buffer and then twice with 50 mM Tris-HCl (pH 7.5), 150 mM NaCl. The captured proteins were then eluted with 200 µL of elution buffer containing 80% 2, 2, 2-trifluoroethanol (Sigma-Aldrich, T63002), 20% PBS with gentle shaking for 30 min at room temperature. The supernatant was collected and dried in a SpeedVac vacuum concentrator (Thermo Scientific).

For Western blot analysis, TUBE pull-down was performed as described above on lysate samples prepared from 293T cells transfected with GFP-OspG or GFP. Beads were then washed three times and the captured proteins were eluted by boiling in 2×SDS-PAGE loading buffer.

## Identification of ubiquitination sites by mass spectrometry

293T cells transfected with plasmids encoding 3×Flag-SEPT9 were harvested and lysed in lysis buffer containing 50 mM Tris-HCl (pH 7.5), 150 mM NaCl, 1% Triton X-100 supplemented with 1% SDS. The cell lysates were denatured for 10 min at 100 °C and then ultrasonicated for 1 min on ice. The supernatants of the cell lysates were collected after centrifugation at 13,000 × *g* for 10 min, and diluted with lysis buffer to a final SDS concentration of 0.1%. The diluted supernatants were incubated with anti-Flag M2 affinity beads for 4 h at 4 °C. The beads were washed three times with lysis buffer, and the samples were eluted and separated on SDS-PAGE for subsequent LC-MS analysis.

## Immunofluorescence and microscopy

Cells were fixed with 4% paraformaldehyde after *S. flexneri* infection, and washed three times with PBS. Cells were then permeabilized with 0.1% Triton X-100 in PBS for 5 min, washed with PBS and blocked with 5% goat serum for 30 min at room temperature. Samples were further incubated with primary antibody in PBS containing 1% BSA and 0.05% Tween-20 overnight at 4 °C, then washed three times with PBS and incubated with secondary antibody for 1 h at room temperature. Cells were washed with PBS, stained with DAPI for 5 min, and then washed three times with PBS. Slides were mounted with antifade mounting medium (Beyotime biotechnology, P0126). Images were captured using a multiphoton microscope (Leica, SP8 DIVE) with an APO 63 × /1.4 oil objective and analyzed using LAS X software (v3.5.6) and Image J software (v1.48).

## Sample preparation for MS analysis

For ubiquitome sample preparation, the dried TUBE pull-down samples were resuspended in 8 M urea, 50 mM ammonium bicarbonate, about 100 µg of proteins per sample were incubated with 5 mM dithiothreitol at 56 °C for 30 min, and alkylated with 25 mM iodoacetamide at room temperature for 1 h in the dark. The samples were diluted with 50 mM ammonium bicarbonate to a final urea concentration of 2 M, and proteins were digested with trypsin (Sigma) overnight at an enzyme-to-protein ratio of 1:50 (w/w) at 37 °C. Finally, the digestion was terminated and acidified by adding TFA to 1%, and the samples were desalted on C18 SepPak solid-phase extraction cartridges (Welch Materials, 00559-11002) and dried. Dried peptide samples were reconstituted and fractionated into twelve fractions on an offline HPLC (SHIMADZU, LC-20AT) using basic pH reversed-phase liquid chromatography (RP-LC) (InertSustain C18 column,

2.1 mm × 50 mm, 5 μm particle size, SHIMADZU). The fractions were dried and stored at −80 °C prior to LC-MS analysis. For LC-MS identification of modified peptides, samples were separated on SDS-PAGE gel, and protein bands of interest were excised and subjected to in-gel digestion.

For in-gel protein digestion, gel bands were excised into 1 mm$^3$ cubes and destained with 50% ACN in 50 mM $NH_4HCO_3$. After dehydration by ACN, samples were incubated with 10 mM dithiothreitol in 10 mM $NH_4HCO_3$ at 56 °C for 30 min, and alkylated with 55 mM iodoacetamide in 10 mM $NH_4HCO_3$ at room temperature for 20 min in darkness. After dehydration by ACN, samples were digested in a buffer containing 1.2 ng/μL trypsin, 50 mM $NH_4HCO_3$ and 10% ACN for 16 h at 37 °C. The tryptic peptides were extracted from gel cubes twice with 50% ACN and 5% formic acid for 20 min at 37 °C. The resulting peptides were pooled and vacuum dried.

### LC-MS analysis
LC-MS analyses of peptide samples were carried out on a hybrid ion trap-Orbitrap mass spectrometer (LTQ Orbitrap Velos, Thermo Scientific) coupled with nanoflow reversed-phase liquid chromatography (EASY-nLC 1200, Thermo Scientific). The capillary column (75 μm × 150 mm) with a laser-pulled electrospray tip (Model P-2000, Sutter instruments) was home-packed with 5 μm, 120 Å Xtimte C18 silicabased particles (Welch, 01710-02100) and run at 300 nL/min with the following mobile phases (A: 97% water, 3% acetonitrile, and 0.1% formic acid; B: 80% acetonitrile, 20% water, and 0.1% formic acid). The LC gradient started at 8% B for 3 min and then was linearly increased to 40% in 40 min. Next, the gradient was quickly ramped to 90% in 2 min and stayed there for 10 min. Eluted peptides from the capillary column were electrosprayed directly onto the mass spectrometer for MS and MS/MS analyses in a data-dependent acquisition mode. One full MS scan (60,000 resolution, $m/z$ 400–1200, $1 \times 10^5$ AGC target, and 500 ms maximal ion injection time) was acquired by the Orbitrap mass analyzer and the 10 most intense ions were selected for fragmentation under collision-induced dissociation (CID) or electron transfer dissociation (ETD). Dynamic exclusion was set with repeat duration of 30 s and exclusion duration of 12 s.

### Label-free protein quantification and statistical analysis
MS/MS spectra were searched against UniProt human protein database (UP000005640) using Mascot software (v2.3.02, Matrix Science). Key search parameters included mass tolerance of 20 ppm for precursor ions and 0.8 Da for MS/MS ions, a maximum of two missed cleavages allowed, carbamidomethylation of Cys (+57.02146 Da) as static modification and Met oxidation (+15.99492 Da) as dynamic modification. Identified peptides were filtered to achieve a false discovery rate (FDR) < 1%. The corresponding peptide peaks were obtained from Thermo Xcalibur (Thermo Fisher Scientific, v2.2).

Spectral counting-based quantification was used in TUBE pull-down and CAND1 pull-down experiments. In CAND1 pull-down experiments ($n = 3$ biological replicates), $P$ values for comparison between nonphosphorylated and phosphorylated CAND1 were derived by Wald test implemented in DESeq2 (R package, v1.34.0). For statistical analysis of TUBE pull-down ubiquitome data ($n = 3$ biological replicates), missing values were replaced by a constant low value to calculate $P$ values. The reported $P$ values for comparison of spectral counts of each protein in *S. flexneri ΔospG: ospG* vs. *S. flexneri ΔospG: vector* infected cells was derived by G-test as described previously[46]. Protein changes in TUBE pull-down ubiquitome were considered statistically significant with $P$ value ≤ 0.05, excluding proteins detected in only one of three replicates or summed spectral counts ≤ 6. Heat map was generated using pheatmap (R package, v1.0.12). GO term enrichment was performed using clusterProfiler (R package, v4.2.2) and GO terms with $P$ values (hypergeometric test implemented in the clusterProfiler package) <0.05 were considered significantly enriched. DAVID functional clustering analysis was performed using a web-based platform at David.ncifcrf.gov. The enriched genes were manually curated so that there were no genes were shared between any two KEGG terms. Volcano plots were created using ggplot2 (R package, v3.4.4). DAVID functional clustering and model images were created using Adobe illustrator 2024 (v28.0).

### Reporting summary
Further information on research design is available in the Nature Portfolio Reporting Summary linked to this article.

## Data availability
The mass spectrometry proteomics data have been deposited to the ProteomeXchange Consortium (https://proteomecentral.proteomexchange.org) via the iProX partner repository with the dataset identifier PXD048447. The link to access the data: https://www.iprox.cn/page/project.html?id=IPX0007864000. The phosphoproteomics, GST-CAND1 pull-down and ubiquitome search results are provided in the Supplementary Information. Source data are provided with this paper.

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

## Acknowledgements

The authors thank the members of the Liu laboratory for the careful review of the manuscript. We are grateful for the constructive discussion with Drs. Feng Shao, Jianyuan Luo, Congying Wu, Jiazhang Qiu, Xing Liu and Yongqun Zhu. This work was financially supported by grants from the National Key Research and Development Program of China (2022YFA1304500 to X.L.), the Natural Science Foundation of China (22174003 and 21974002 to X.L., 32370185 to J.F.), the National Institutes of Health (R01AI127465 and R01GM126296 to Z.Q.L.), State Key Laboratory of Proteomics, Peking University Medicine Seed Fund for Interdisciplinary Research and the Fundamental Research Funds for the Central Universities.

## Author contributions

X.L., Z.Q.L., W.X. and J.F. conceived and designed the study. W.X. and J.F. performed most experiments. Y.Z., P.S.B., N.Z., S.O. and Y.B.Z. provided technical support and advice on bacterial infection and OspG/CAND1 biology. Q.Z., Y.Y. and Y.W. helped with the bacterial infection and plasmid construction. C.L. helped with the CAND1-cullin interaction assay. Z.T. assisted with the ubiquitome analysis. X.L., Z.Q.L., W.X. and J.F. analyzed the data and wrote the manuscript with input from all authors.

## Competing interests

The authors declare no competing interests.
