## [Peer Review File · Nature Communications]

The *Shigella* kinase effector OspG modulates host ubiquitin signaling to escape septin-cage entrapmentREVIEWER COMMENTS

Reviewer #1 (Remarks to the Author):

OspG is a *Shigella* effector previously shown to bind to the ubiquitin conjugation machinery (E2-Ub) to activate its kinase activity; however cellular role of OspG is not fully known. The authors report that CAND1 is target of OspG (using phosphoproteomics) and that CAND1 controls assembly of cullin-RING ubiquitin ligases (CRLs). CAND1 phosphorylation weakens interactions with CRLs, changing the ubiquitin landscape of cells. Ubiquitome profiling suggests that OspG promotes ubiquitination of the septin cytoskeleton. Finally, the authors propose that ubiquitination of septins by OspG inhibits assembly of cages surrounding *Shigella*.

The authors discover interesting and novel OspG biology potentially of great interest to the field. That bacteria may encode for an effector to escape from septin cage entrapment would propel the role of septins in cell-autonomous immunity. However further experiments are required to support the authors conclusions.

In vitro ubiquitination experiments (as in Figure 4) are challenging to obtain reliable, consistent results. In this case, authors report K6, K27, K33 mediated ubiquitination (notoriously non-specific chain types) and all reductions reported by WB in Figure 4A/B are very slight (especially considering cells are overexpressing OspG). Considering this, results in Figure 4 are not compelling. How often were experiments performed? Provide molecular markers. Use alternative approaches (such as reciprocal pulldowns) so that robust conclusions can be drawn.

Figure 4E (cellular impact) is weak. How often were experiments performed? Provide molecular markers. Complement with alternative approaches (such as microscopy testing which septin assemblies are affected). Test for cullin-septin interactions during OspG over expression or *Shigella* infection.

What is the role of OspG? From the literature, there are mutants available to more fully test the role of OspG kinase activity versus E2 binding / inhibiting ability in functions being reported here.

Figure 5 is currently difficult to conclude from. These cells are suffering / dying from massive *Shigella* infection. Lower the MOI and increase time of infection, ie perform experiments concordant with other septin caging literature more reliably capturing septin caging (eg Mostowy et al 2010).

Figure 5C: Perform this experiment so that septin caging is being significantly detected in your WT cells.

Figure 5D: Label for both ubiquitin (FK2) and septin caging. Test for recruitment of CAND1, cullin; do we expect these other markers to also be at the septin cage? Considering that septins form both hexamers (SEPT7+ve, SEPT9-ve) and octamers (SEPT+ve, SEPT9+ve), test for SEPT7 cages. Are SEPT7 cages compromised +/- OspG?

Figure 5E: Why does Y axis change to cells with % caging? I assume this will not be required once reliable septin cage counts are being obtained.

Considering that OspG is under the control of MxiE, test the role of MxiE mutant to support results being obtained using OspG.

SEPT9 is ubiquitinated, but how does ubiquitination impair septin assembly and cage entrapment? Does ubiquitination inhibit assembly into filaments or into cage-like structures? Are septin filaments compromised in OspG transfected cells? Are septins visibly ubiquitinated inside cells? Are other septin functions (besides cage assembly) impaired by ubiquitination?

Which SEPT9 isoform is under investigation? This is important considering the different actin vs microtubule binding domains (see literature by Mavrakis, Spiliotis)? Which isoform is your SEPT9 antibody recognising?

Reviewer #2 (Remarks to the Author):

The manuscript „The Shigella kinase effector OspG hijacks host ubiquitin signaling to escape septin-cage entrapment” by Xian et al. describes the mechanism of action of OspG, a major effector protein of the intracellular pathogen *Shigella flexneri*. Starting with unbiased phosphoproteomics, the authors determined VAND1 as a major target of the OSp1 kinase activity, proceeded to show the direct interaction between the enzyme and substrate and functional relationship involving the control of the ubiquitin machinery that in turn leads to the manipulation of host septins, a class of cytoskeletal proteins that form a cage-like structure trapping cytosolic bacteria and therefore helping *Shigella* evade the host defense system. The well-designed sequence of experiments using a variety of methods resulted in a good conclusion and mechanistic explanation. However, before the manuscript is accepted for publication, this reviewer feels there are some concerns that need to be addressed.

Detailed comments:

1. First and foremost, the raw mass spectrometry data have to be deposited in a publicly accessible database (such as ProteomeXchange) – data availability is a requirement for proteomics experiments. Similarly, the processed data should be presented as supplemental tables. Without these two components, essential evidence supporting all the experiment is missing and therefore impossible to assess the quality of the experiments that form the base of the conclusions.
2. Page 6, last sentence: are there any other predicted targets of OspG kinase activity? What about the recognition sequence?
3. Page 7. Lines 142-144: why was only Cul1 validated by immunoblotting and not the other cullins?
4. Page 7, Lines 145-146: the phosphorylation of CAND1 co-expressed on *E. coli* (presumably with OspG?) may not be physiologically relevant, so the additional sites discovered there may not be significant. What is the control showing that they are not phosphorylated in CAND1 expressed without OspG in *E. coli*? The authors suggest the importance of these sites based on evolutionary conservation in other species. Have these sites been shown to be phosphorylated on other species? By what kinases? Is OspG shown to be an ortholog of any other kinases? This is a weak part of the paper and may be unnecessary, or it should be developed.
5. Page 8, line 153. Following up on that: “The mutation of most sites had limited impact” – is there a supplemental figure showing this limited impact? Also, mutagenesis is a great tool, but not necessarily perfect.

Reviewer #3 (Remarks to the Author):

Xian et al. identified CAND1 as a kinase substrate of the *Shigella* effector OspG using a global phosphoproteomic analysis and some biochemical assays. Phosphorylation of CAND1 subsequently disrupted its interaction with Cul1, which led to impair the SCF complexes. Moreover, a global ubiquitination profiling identified septins that were affected by OspG-induced remodeling of the cellular ubiquitome profile. Finally, the authors showed that the ubiquitination of septins prevents the assembly of septin-cages during infection. However, there are some technical issues to be addressed.

1. Line 31: Why had targets of OspG remained elusive due to extensive efforts? What the authors did in this study is a typical, straightforward, and differential phosphoproteomic analysis between OspG WT vs mutant. I just wondered why none of the studies had identified the OspG targets. Any specific reasons?
2. Fig. 1B: Why spectral count was used for quantification of phosphopeptides? First, some studies showed that peak area or intensity-based label-free quantification (e.g., MaxLFQ [PMID: 24942700]) outperforms the spectral count. Second, typical phosphoproteome analyses rely on spectral counts of “single or a few (phospho)peptide”, in contrast to whole proteome analyses in which some distinct peptides derived from the same protein are quantified. Therefore, I suspect

that quantification accuracy using spectral count (Fig. 1B) is not as good as peak area-based LFQ.

3. Fig. 1B: Along the same line, there is no information about the reproducibility of replicates and corrected p-values. Ideally, a volcano plot (x-axis: log₂ FC, y-axis: corrected p-value) would be more informative than Fig. 1B.

4. Figs. 1, 2, 3: Please provide Excel tables that describe sequences of phosphopeptides identified, phosphorylation sites, quantification values, scores, and other associated information. I also could not find such tables about the ubiquitination profiling (Figs. 2, 3).

5. Have the raw data and results files been deposited on a proteomic server? Please deposit them so that reviewers and readers can check and reanalyze them.

6. Line 112: Can AlphaFold2 further support a complex of CAND1-OspG-Ub (vs CAND1-OspG only)? This may provide additional and structural evidence that CAND1 is a "direct" target of OspG.

7. Line 122 (Fig. 1G): Based on the intensities of the non-phosphorylated peptide and phosphorylated peptide, the phosphorylation stoichiometry in the infected cells seems to be very low ~1%. Is this reasonable? Please also mention how much percentage of total cells were infected by Shigella.

8. Line 148: Can the AlphaFold2-based predicted structure of OspG tell that it is a dual serine and tyrosine kinase?

9. Fig 3A: A more proper control would be an OspG mutant rather than a vector only. Why was not the OspG mutant used?

10. Fig. 3C: The replicates did not show consistent results, and the quantification values (the heatmap) varied among the replicates. This indicates that the results are not reproducible. The authors should comment on that. Regarding the comment 2, the authors should use peak area-based LFQ rather than the spectral counting-based method.

11. Line 380: The method part about phosphoproteomics was not well-documented compared to other proteomic methods. Please describe it more clearly regarding reduction/alkylation, lysc/trypsin treatment, and the way to enrich phosphopeptides, etc.

12. Sholtz et al. revealed that Enteropathogenic E. coli and Shigella induced phosphorylation of Ser30 on septin-9 (SEPT9) for bacterial adherence and cell death [PMID: 25944883]. Does OspG also target Ser30 on septin-9 in addition to CAND1? Discussion with citation of that paper would be needed.

We are very grateful to all three reviewers and the editor for your insightful comments and helpful suggestions. We have carried out further investigations and made substantial improvements to the manuscript. Please kindly find point-by-point responses below.

REVIEWER COMMENTS

Reviewer #1:

OspG is a *Shigella* effector previously shown to bind to the ubiquitin conjugation machinery (E2-Ub) to activate its kinase activity; however cellular role of OspG is not fully known. The authors report that CAND1 is target of OspG (using phosphoproteomics) and that CAND1 controls assembly of cullin-RING ubiquitin ligases (CRLs). CAND1 phosphorylation weakens interactions with CRLs, changing the ubiquitin landscape of cells. Ubiquitome profiling suggests that OspG promotes ubiquitination of the septin cytoskeleton. Finally, the authors propose that ubiquitination of septins by OspG inhibits assembly of cages surrounding *Shigella*.

The authors discover interesting and novel OspG biology potentially of great interest to the field. That bacteria may encode for an effector to escape from septin cage entrapment would propel the role of septins in cell-autonomous immunity. However further experiments are required to support the authors conclusions.

In vitro ubiquitination experiments (as in Figure 4) are challenging to obtain reliable, consistent results. In this case, authors report K6, K27, K33 mediated ubiquitination (notoriously non-specific chain types) and all reductions reported by WB in Figure 4A/B are very slight (especially considering cells are overexpressing OspG). Considering this, results in Figure 4 are not compelling. How often were experiments performed? Provide molecular markers. Use alternative approaches (such as reciprocal pulldowns) so that robust conclusions can be drawn.

Response: We do appreciate very much your constructive comments as well as considering our work being interesting and novel. We agree that ubiquitination experiments can be challenging...That being said, the data we presented in the original Fig. 4 (now Fig. 5) were fairly reproducible. We indeed repeated these experiments multiple times (>3) to figure out exactly which septins (out of a panel including SEPT2/6/7/9/10/11) underwent ubiquitination in an OspG-dependent manner. For individual ubiquitinated septins, our take is that only a fraction of the total pool is modified. Therefore, the ubiquitination signals in immunoblots are in general rather moderate (and weak in some cases). One possible explanation is that OspG does NOT directly target/modify septins (via phosphorylation of CAND1 instead). From another perspective, low rates of effector-mediated modifications (in particular during infection) can be beneficial for pathogens in that fully modifying host targets can often be detrimental to host cells. In infected cells, it would make sense to target only the fraction of septins that actually form the cages. As seen in the immunostaining, septins actually forming cages account for a rather small portion of total protein pools.

As you suggested, we conducted reciprocal pulldowns (if we understand correctly) to further

support our findings. Basically, TUBE-based pulldown of all ubiquitinated proteins were performed and then blotted with specific antibodies to monitor the presence of SEPT7 or SEPT9. Indeed such experiments confirmed the ubiquitination of SEPT9 but not SEPT7 in an OspG-dependent manner (Fig. S3A and see lines 210-218 in the text for more description). In addition, these blots clearly indicate that only a small fraction of total SEPT9 was ubiquitinated. We think that as long as the small portion of cage-forming septins is modified, it fulfills the duty of breaking down cages. In an independent study (PMID: 34739333), low levels of K63-linked ubiquitination of G3BP1 are shown to be capable of exerting its function as well (i.e., mediate stress granule disassembly) in cells. We have now provided molecular markers in all WB images.

Figure 4E (cellular impact) is weak. How often were experiments performed? Provide molecular markers. Complement with alternative approaches (such as microscopy testing which septin assemblies are affected). Test for cullin-septin interactions during OspG over expression or Shigella infection.

Response: We agree with you regarding the original Fig 4E (now as Fig 5D). The actual impact is presumably larger than it appears in the figure, partially due to the overexpression of SEPT2/6/7 in cells. In addition, as we discussed above only a portion of septins undergoes ubiquitination. Therefore, the ultimate readout on septin-septin interactions was not dramatic as one would like to see. We have tried microscopy testing which didn't yield positive results. To facilitate the observation of such differences, we included the relative abundance of Western bands (see Fig. 5D). At least for SEPT7-SEPT9 interaction, the difference is definitely there. That being said, a clean experiment that fits such purpose would be in-vitro assays with fully ubiquitinated proteins.

As suggested, we also tested potential interactions of septins with cullins or CAND1 (see the figure below). In most cases, we found rather weak associations of these proteins (if there is any). Such findings are not something unexpected, given that the F-box proteins (FBPs), but not cullins (as a scaffold), are responsible for direct substrate recruitment in the SCF complex. Of note, for the same reason (i.e., weak E3-substrate interactions) it is often challenging to identify substrates for ubiquitin ligases (PMID: 22962057).

Immunoblots of SEPT-cullins/CAND1 interactions. 293T cells transfected with Flag-SEPT either alone or with GFP-OspG. Immunoprecipitated samples were further probed with indicated antibodies.

What is the role of OspG? From the literature, there are mutants available to more fully test the role of OspG kinase activity versus E2 binding / inhibiting ability in functions being reported here.

Response: Indeed, previous reports have shown that Lys-53 of OspG is the catalytic residue that anchors and orients ATP. Mutation of two leucine residues (L190D/L191D) abolishes OspG binding to ubiquitin (PMID: 23469023) and OspG LR/PE mutant is incapable of binding to E2~Ub (PMID: 24856362). We constructed these OspG variants and found that disruption of either its kinase activity or association with E2/Ub renders the effector unable to phosphorylate CAND1 (Fig. 1D). Nevertheless, in-vitro GST pull-down assays showed that these OspG mutants can still bind to CAND1 as efficiently as the wild-type effector (Fig. S1A). Therefore, it seems that OspG requires the association with E2/Ub for kinase activation but not CAND1 targeting.

In addition, decreased levels of SEPT9 ubiquitination were observed for these mutants (Fig 5C), and even lower levels were seen for the double mutants (LL/DD K53A and LR/PE K53A). Together, these data suggest that E2/Ub association is required for CAND1 phosphorylation/SEPT9 ubiquitination by stimulating OspG kinase activity.

Figure 5 is currently difficult to conclude from. These cells are suffering / dying from massive Shigella infection. Lower the MOI and increase time of infection, ie perform experiments concordant with other septin caging literature more reliably capturing septin caging (eg Mostowy et al 2010).

Response: Thank you for the suggestion. We have redone this experiment under the conditions reported in the literature (e.g., lower MOI, infection with bacteria grew to OD₆₀₀=0.6) (Fig. 6A and 6C).

Figure 5C: Perform this experiment so that septin caging is being significantly detected in your WT cells.

Response: As suggested by the reviewer, we have optimized some experimental conditions as previously described (Mostowy S, et al. 2010; Lobato-Márquez D, et al. 2023) and managed to get higher percentage of septin caging in WT cells (Fig. 6B and Fig. 6D).

Figure 5D: Label for both ubiquitin (FK2) and septin caging. Test for recruitment of CAND1, cullin; do we expect these other markers to also be at the septin cage? Considering that septins form both hexamers (SEPT7+ve, SEPT9-ve) and octamers (SEPT+ve, SEPT9+ve), test for SEPT7 cages. Are SEPT7 cages compromised +/- OspG?

Response: As suggested, we have stained both ubiquitin (FK2) and septin caging. We were able to find significant co-localization of FK2 signals with some deformed cages (likely corresponding to those cages under destruction by OspG-mediated ubiquitination) in cells infected by WT bacteria but not the *ospG*-deletion mutant (Fig. S5B and see lines 266-271 in the text). Given the weak association of the E3 complex with septins, we didn't proceed with the staining of CAND1 or cullins in cells. We have stained SEPT7 cages and similar data were observed (i.e., inhibition of cage assembly by OspG) (Fig. 6B and Fig. S5A).

Figure 5E: Why does Y axis change to cells with % caging? I assume this will not be required once reliable septin cage counts are being obtained.

Response: Agree, we have changed the Y axis accordingly (see the new Fig. 6D).

Considering that OspG is under the control of MxiE, test the role of MxiE mutant to support results being obtained using OspG.

Response: We have constructed the $\Delta mxiE$ mutant via a CRISPR-Cas9 system and tested cage formation in infected cells. We found that cage assembly in cells infected by the $\Delta mxiE$ mutant was not significantly different from that in WT infection (see the figure below). To further explain such observations, we measured the protein levels of OspG (and MxiE) in different strains (shown in the figure below). Notably, OspG was robustly expressed in the $\Delta mxiE$ mutant, though at a slightly lower level than that of WT bacteria.

(A) Schematic diagram of generating the *mxIE* knockout strain. (B) Genome typing of *S. flexneri* by PCR.

(A) HeLa cells were infected with *S. flexneri* at a MOI of 10 for 2 h, fixed and stained with anti-SEPT9 or anti-SEPT7 antibody for quantitative microscopy. Data represent the mean % \pm SEM of *S. flexneri* inside septin-cages. Data are representative of three biologically independent experiments. (B) Quantification of the relative abundance of OspG and MxiE of indicated strains by proteomic analysis.

SEPT9 is ubiquitinated, but how does ubiquitination impair septin assembly and cage entrapment? Does ubiquitination inhibit assembly into filaments or into cage-like structures? Are septin filaments compromised in OspG transfected cells? Are septins visibly ubiquitinated inside cells? Are other septin functions (besides cage assembly) impaired by ubiquitination?

Response: Good question. We have mapped the ubiquitination sites to both N-terminal and C-terminal domains of SEPT9 but not the GTPase domain (Fig. S3E). Notably, its N-terminal domain interacts directly with actin filaments and microtubules. To address the question how ubiquitination impairs septin function, in-vitro filament or cage assembly assays would be desired. To do this, it is necessary to obtain fully modified septins and compare them with the unmodified controls. Given OspG mediates ubiquitination of septins in an indirect manner, it would be challenging to conduct such in-vitro biochemical experiments. That being said, we have found other effectors directly modify septins (not ubiquitination) with high efficiency under transfection conditions (unpublished data). Co-expression of these effectors and septins in *E. coli* led to the purification of highly modified septins. Therefore, we will be able to assess the impact of modification by using in vitro assembly assays in another study.

As suggested, the discovery of septin modifications opens up more intriguing questions in septin biology that await further investigations (some of which may be beyond the scope of this study). We have made some attempts to examine potential disruption of septin filaments in OspG-expressing cells. Thus far, we haven't seen any positive outcome...As we stated earlier, this may be partially due to low rates of OspG-mediated ubiquitination. For the same reason, visualization of septin ubiquitination in cells would be technically challenging at this stage as well.

Which SEPT9 isoform is under investigation? This is important considering the different actin vs microtubule binding domains (see literature by Mavrakis, Spiliotis)? Which isoform is your SEPT9 antibody recognising?

Response: We used isoform a in this study (also described in the revised method section). The immunogen sequence of our SEPT9 antibody is “LGVKNSEPSARHVDSLQSRSPKASLRRVELSGPKAAEPVSRRELSIDISSKQVENAGAIGPSRFGLKRAEVLGHKTPEPAPRTEITIVKPQESA”, which suggests it can recognize several isoforms including a, b, c, and g.

Reviewer #2:

The manuscript “The Shigella kinase effector OspG hijacks host ubiquitin signaling to escape septin-cage entrapment” by Xian et al. describes the mechanism of action of OspG, a major effector protein of the intracellular pathogen *Shigella flexneri*. Starting with unbiased phosphoproteomics, the authors determined CAND1 as a major target of the OspG kinase activity, proceeded to show the direct interaction between the enzyme and substrate and functional relationship involving the control of the ubiquitin machinery that in turn leads to the manipulation of host septins, a class of cytoskeletal proteins that form a cage-like structure trapping cytosolic bacteria and therefore helping *Shigella* evade the host defense system. The well-designed sequence of experiments using a variety of methods resulted in a good conclusion and mechanistic explanation. However, before the manuscript is accepted for publication, this reviewer feels there are some concerns that need to be addressed.

Response: We do appreciate your positive comments regarding our work.

Detailed comments:

1. First and foremost, the raw mass spectrometry data have to be deposited in a publicly accessible database (such as ProteomeXchange) – data availability is a requirement for proteomics experiments. Similarly, the processed data should be presented as supplemental tables. Without these two components, essential evidence supporting all the experiment is missing and therefore impossible to assess the quality of the experiments that form the base of the conclusions.

Response: Thanks for the reminder. We have deposited raw data RAW mass spec data (i.e., phosphoproteome, ubiquitome and etc.) to the ProteomeXchange Consortium (<https://proteomecentral.proteomexchange.org>) via the iProX partner repository with the dataset identifier PXD048447. The link to access the data: <https://www.iprox.cn/page/project.html?id=IPX0007864000>. The processed data (mostly in Excel format) have also been included as supplemental materials (see Supplemental Table 1-3).

2. Page 6, last sentence: are there any other predicted targets of OspG kinase activity? What about the recognition sequence?

Response: Prior to our phosphoproteome analysis, we thought we may end up with too many candidates from such a high-throughput screen. In the end, we were a bit surprised that we had only one or two hits to follow up...In addition to the CAND1 peptide, the other phosphorylated peptide (AAQAPS**S**FQLLYDLK) that stands out comes from C1TC, a metabolic enzyme. Few candidate targets of OspG may suggest that this bacterial kinase is highly specific. Nonetheless, we have also seen some bacterial enzymes (kinases and other enzymes as well) with a broad spectrum of host substrates. Anyway, we have tried to employ pLogo to determine any potential motif OspG may recognize with the pool of peptides that were statistically different in phosphoproteomics ($p < 0.05$, fold change > 3), yet such exercise was not successful or informative perhaps due to the small sample size.

Consensus motif analysis by pLogo.

3. Page 7. Lines 142-144: why was only Cull1 validated by immunoblotting and not the other cullins?

Response: Among all cullin-RING ubiquitin ligases, the SCF (Skp1-Cull1-F-box proteins) complex is arguably the best characterized one. In addition, thus far most structural studies of cullin-CAND1 complex used Cull1 as well. Therefore, we prioritized Cull1 when we evaluated the potential impact of phosphorylation on CAND1-cullin interactions. That being said, when we examined the interactions of CAND1 S376D with cullins we also included other identified ones including Cul4A/4B/5 (see Fig S2C).

4. Page 7, Lines 145-146: the phosphorylation of CAND1 co-expressed on E.coli (presumably with OspG?) may not be physiologically relevant, so the additional sites discovered there may not be significant. What is the control showing that they are not phosphorylated in CAND1 expressed without OspG in E. coli? The authors suggest the importance of these sites based on evolutionary conservation in other species. Have these sites been shown to be phosphorylated on other species? By what kinases? Is OspG shown to be an ortholog of any other kinases? This is a weak part of the paper and may be unnecessary, or it should be developed.

Response: Yes, the additional sites were identified from CAND1 co-expressed with OspG in *E. coli*, and we agree that their physiological relevance may be under debate. Having said that, we included multiple rigorous controls (see the figure below) to confirm the identification of these sites. Despite being a master regulator of protein ubiquitination, CAND1 phosphorylation (or any other modifications) has not been functionally studied on any species. Therefore, any findings on new modification sites of CAND1 may be of great interest to the ubiquitination field. Given the uncertainty about their physiological relevance, we decided to tone down this part of the data. We have added the following statement “whether they are also phosphorylated in eukaryotic cells may warrant further investigations.” (see lines 155-156 in the text).

OspG does have a homolog (NleH1) from pathogenic *E. coli*, and we have included this effector as a control in the revision (see Fig. 1H). NleH1 modifies another important host protein

(work in progress in our group) but not CAND1.

Extracted ion chromatograms of Ser376, Ser558, Tyr784, Tyr980-phosphorylated peptides and a control peptide (LTLIDPETLLPR) from indicated GST-CAND1 samples. *nonspecific peaks.

5. Page 8, line 153. Following up on that: “The mutation of most sites had limited impact” – is there a supplemental figure showing this limited impact? Also, mutagenesis is a great tool, but not necessarily perfect.

Response: We agree that mutagenesis is not necessarily perfect, though it is a straightforward approach. We have examined the structure of CAND1-Cul1 complex and the identified modification sites including S376D are not located at the interacting interface. For the one (i.e., S376D) that does have a dramatic impact, it likely involves an overall conformational change of the protein. We are currently collaborating on solving the structure of phosphoCAND1. In the meantime, to avoid potential confusion with “this limited impact”, we have rephrased the original text to “Among those tested mutants, the CAND1 phosphomimetic variant S376D almost lost its ability to interact with endogenous Cul1 (Fig. 2E)” (see lines 161-162).

Reviewer #3:

Xian et al. identified CAND1 as a kinase substrate of the Shigella effector OspG using a global phosphoproteomic analysis and some biochemical assays. Phosphorylation of CAND1 subsequently disrupted its interaction with Cull1, which led to impair the SCF complexes. Moreover, a global ubiquitination profiling identified septins that were affected by OspG-induced remodeling of the cellular ubiquitome profile. Finally, the authors showed that the ubiquitination of septins prevents the assembly of septin-cages during infection. However, there are some technical issues to be addressed.

1. Line 31: Why had targets of OspG remained elusive due to extensive efforts? What the authors did in this study is a typical, straightforward, and differential phosphoproteomic analysis between OspG WT vs mutant. I just wondered why none of the studies had identified the OspG targets. Any specific reasons?

Response: You actually raised a very legitimate question...Indeed, we had the same curiosity right before this project was initiated about a decade ago (unfortunately a long time...). At that time, it came almost natural to us that we should use phosphoproteomics to screen for kinase substrates. That being said, such a spontaneous reaction for mass spectrometrists may not be that straightforward to most microbiologists. Even when we wrote this manuscript and share with our colleagues in microbiology, we were kindly reminded that it may not be taken well to start the work with a proteomic screen...Perhaps starting with a genetic screen for phenotypes would be better...Indeed most of previous studies on OspG focused on its role in regulating NF- κ B pathway (which we think can be a minor effect). Of course, those studies have not been able to nail down its phosphorylated targets.

2. Fig. 1B: Why spectral count was used for quantification of phosphopeptides? First, some studies showed that peak area or intensity-based label-free quantification (e.g., MaxLFQ [PMID: 24942700]) outperforms the spectral count. Second, typical phosphoproteome analyses rely on spectral counts of “single or a few (phospho)peptide”, in contrast to whole proteome analyses in which some distinct peptides derived from the same protein are quantified. Therefore, I suspect that quantification accuracy using spectral count (Fig. 1B) is not as good as peak area-based LFQ.

Response: We certainly agree with you that peak area or intensity in general is more accurate than spectral counting, which indeed seems to be quite outdated in this day and age...In our experience, the old recipe may be less accurate or sensitive in picking up potential differences, yet the relative conservativeness may benefit us in that if spectral counting identifies a trend, it got to be right! In addition, in most cases we tend to pick the most differed proteins for follow-up studies. For those guys, one can hardly miss them even by eyeballing (basically staring at the counts) without sophisticated statistics...Last but not the least, the old method is just QUICK. That being said, we would often use peak area for publication purpose, though the initial discovery was made by counts. For this particular phosphoproteome dataset (collected many years ago), unfortunately LC-MS experiments were carried out with a LTQ Velos Pro instrument. The low-resolution data preclude us from using LFQ quantification. For that, we do appreciate very much in advance for

your understanding.

3. Fig. 1B: Along the same line, there is no information about the reproducibility of replicates and corrected p-values. Ideally, a volcano plot (x-axis: log₂ FC, y-axis: corrected p-value) would be more informative than Fig. 1B.

Response: Yes, Fig. 1B is not a standard volcano plot. For LFQ quantification data, we would usually plot log₂ FC as x axis and p-value as y axis. For phosphoproteome, we conducted LC-MS analyses of four replicates (see lines 80-83). As we knew from the very beginning that we will have to finish up the functional studies of identified targets, we got to be very confident with the reproducibility of our proteomic data. Otherwise, the price paid for follow-up investigations would simply be too high...

4. Figs. 1, 2, 3: Please provide Excel tables that describe sequences of phosphopeptides identified, phosphorylation sites, quantification values, scores, and other associated information. I also could not find such tables about the ubiquitination profiling (Figs. 2, 3).

Response: Thanks for the suggestions and we have included those information as Supplemental Table 1-3.

5. Have the raw data and results files been deposited on a proteomic server? Please deposit them so that reviewers and readers can check and reanalyze them.

Response: We have deposited raw data to the ProteomeXchange Consortium (<https://proteomecentral.proteomexchange.org>) via the iProX partner repository with the dataset identifier PXD048447. The link to access the data: <https://www.iprox.cn/page/project.html?id=IPX0007864000>.

6. Line 112: Can AlphaFold2 further support a complex of CAND1-OspG-Ub (vs CAND1-OspG only)? This may provide additional and structural evidence that CAND1 is a “direct” target of OspG.

Response: We have made some attempts by using AlphaFold2 to predict the structure of CAND1-OspG-Ub (see the figure below). Yet we were advised by structural biologists that current prediction of three-subunit complex structure may not be that accurate. AlphaFold2 works best for single proteins.

(A) The AlphaFold2-predicted structure of the CAND1-OspG-Ub complex. (B) Sequence coverage scores in the structure prediction. (C) The pLDDT scores of the five predicted models.

7. Line 122 (Fig. 1G): Based on the intensities of the non-phosphorylated peptide and phosphorylated peptide, the phosphorylation stoichiometry in the infected cells seems to be very low ~1%. Is this reasonable? Please also mention how much percentage of total cells were infected by *Shigella*.

Response: Indeed, the modification rates during infection are generally quite low (1% or less)...which is based on our experience in the last decade working with effector-mediated modifications (PMID: 21822290 (phosphocholination), 30323948 (glucosylation), 29795342 (phosphoribosyl-linked serine ubiquitination), 34671164 (ADP-riboxanation)). The low rates are largely due to the minimal amounts of effectors secreted by bacterial pathogens during infection. Our understanding is that pathogens tend to target/modify their host substrates in a very precise manner to avoid large perturbations in host cells. After all, they want to live together with the host. Please also see our response to the first comment raised by Reviewer 1.

Under the conditions we used for *S. flexneri* infection, typically all host cells are eventually infected by bacteria (as we transformed *Shigella* with a plasmid expressing afimbrial adhesin to increase infection rates).

8. Line 148: Can the AlphaFold2-based predicted structure of OspG tell that it is a dual serine and tyrosine kinase?

Response: Indeed the crystal structure of OspG has been solved for many years (PMID: 24446487, 24856362). At the very moment we came to know that it is a dual serine and tyrosine kinase, honestly we were quite astonished. Such information was not predicted or suggested in the previous structural studies.

9. Fig 3A: A more proper control would be an OspG mutant rather than a vector only. Why was not the OspG mutant used?

Response: You are definitely right. In principle, the use of a catalytic mutant would be a better control. Well, the less “perfect” vector control turned out to serve us better for the purpose of teasing out septins in the ubiquitome profiling (though we did NOT plan on this for sure). As shown by the Western data (Fig. 5A), the differences in ubiquitination levels are the largest between the vector and wild-type effector.

10. Fig. 3C: The replicates did not show consistent results, and the quantification values (the heatmap) varied among the replicates. This indicates that the results are not reproducible. The authors should comment on that. Regarding the comment 2, the authors should use peak area-based LFQ rather than the spectral counting-based method.

Response: We agree that reproducibility of the ubiquitome data (e.g., the original Fig. 3C (now Fig. 4A)) did not look as good as we would like to be. Our current protocol of ubiquitome profiling is quite lengthy and tedious. The whole process was not as pleasant at all as the phosphoproteome analysis. Overall, we hardly observed any dramatic differences as we typically encounter in phosphoproteome datasets. We think this is, at least in part, due to the fact that a major role of ubiquitination is for protein degradation. Therefore, some ubiquitinated proteins may be degraded prior to their enrichment for quantitative proteomics (PMID: 22962057), leading to underestimated fold changes.

As suggested, we made the attempts to redo the quantification of ubiquitome datasets by MaxQuant LFQ approach. As shown below, the updated heatmap seems to look better in terms of reproducibility (Fig. 4A). We have also made a volcano plot (shown below) for your reference with septins highlighted.

(A) Heatmap illustrating statistically significant changes in the ubiquitome of *S. flexneri* infected cells.

(B) A volcano plot illustrating statistically significant changes in the ubiquitome of *S. flexneri* infected cells. Candidate substrates (i.e., septins) are labeled with their names in the graph.

11. Line 380: The method part about phosphoproteomics was not well-documented compared to other proteomic methods. Please describe it more clearly regarding reduction/alkylation, lysc/trypsin treatment, and the way to enrich phosphopeptides, etc.

Response: We have significantly expanded this section and provided most experimental details as suggested (see lines 419-439).

12. Sholtz et al. revealed that Enteropathogenic *E. coli* and *Shigella* induced phosphorylation of Ser30 on septin-9 (SEPT9) for bacterial adherence and cell death [PMID: 25944883]. Does OspG also target Ser30 on septin-9 in addition to CAND1? Discussion with citation of that paper would be needed.

Response: We do appreciate your comments. It is GOOD and interesting to know that SEPT9 undergoes phosphorylation as well. We measured phosphorylation of Ser30 and Ser85 on SEPT9 in OspG-expressing cells versus controls (Fig. S4C-D). Our data indicate that an endogenous host kinase, but not OspG, likely phosphorylates SEPT9. We have discussed in the text (see lines 307-312). Nonetheless, the functional role(s) of SEPT9 phosphorylation would certainly warrant further investigations and likely add intriguing new dimensions to our understanding of septin biology.

REVIEWERS' COMMENTS

Reviewer #1 (Remarks to the Author):

The authors have addressed all of my comments and manuscript is significantly improved. A lot of work was done for revision (eg new mutants constructed), and authors are honest / have explanations when results not 'perfect'.

This is an exciting report and will be of great interest to researchers in the field of septin biology, Shigella infection, ubiquitin signaling, and cellular immunity. I look forward to the impact this discovery will have.

Minor comments:

Before moving to publication, please include mxiE data (so readers in the field can appreciate these results), and make clear in the manuscript when the hyper-invasive strain of Shigella (AfaE strain) is being used.

I would have preferred high resolution microscopy for cage / Ub data, but agree with interpretation that Ub is recruited to deformed cages under destruction.

Reviewer #2 (Remarks to the Author):

The authors have addressed my concerns in a satisfactory manner. Thank you!

Reviewer #3 (Remarks to the Author):

The authors have addressed most of the concerns, so I support publication in Nature Communications.

REVIEWER COMMENTS

Reviewer #1:

The authors have addressed all of my comments and manuscript is significantly improved. A lot of work was done for revision (eg new mutants constructed), and authors are honest / have explanations when results not 'perfect'.

This is an exciting report and will be of great interest to researchers in the field of septin biology, *Shigella* infection, ubiquitin signaling, and cellular immunity. I look forward to the impact this discovery will have.

Response: Thank you very much for your appreciation as well as support!

Minor comments:

Before moving to publication, please include mxiE data (so readers in the field can appreciate these results), and make clear in the manuscript when the hyper-invasive strain of *Shigella* (AfaE strain) is being used.

Response: As suggested, we have now presented the MxiE data in both Fig. 6B and Supplementary Fig. 5B. The use of the hyper-invasive strain has also been clearly stated in the method section (see lines 388 to 389).

I would have preferred high resolution microscopy for cage / Ub data, but agree with interpretation that Ub is recruited to deformed cages under destruction.

Response: Thank you for the agreement. We have tried the Leica Lightning 3D deconvolution method to capture better images (data presented in Supplementary Fig. 6B), though the improvement in resolution seems to be marginal (without a better microscope).

Reviewer #2:

The authors have addressed my concerns in a satisfactory manner. Thank you!

Reviewer #3:

The authors have addressed most of the concerns, so I support publication in Nature Communications.